# Time-MMD: Multi-Domain Multimodal Dataset for Time Series Analysis

**Haoxin Liu**[†][*] **Shangqing Xu**[†], **Zhiyuan Zhao**[†], **Lingkai Kong**[†],
**Harshavardhan Kamarthi**[†], **Aditya B. Sasanur**[†], **Megha Sharma**[†],
**Jiaming Cui**[†], **Qingsong Wen**[§], **Chao Zhang**[†], **B. Aditya Prakash**[†][*]

[†]Georgia Institute of Technology     [§]Squirrel AI

## Abstract

Time series data are ubiquitous across a wide range of real-world domains. While real-world time series analysis (TSA) requires human experts to integrate numerical series data with multimodal domain-specific knowledge, most existing TSA models rely solely on numerical data, overlooking the significance of information beyond numerical series. This oversight is due to the untapped potential of textual series data and the absence of a comprehensive, high-quality multimodal dataset. To overcome this obstacle, we introduce Time-MMD, the first multi-domain, multimodal time series dataset covering 9 primary data domains. Time-MMD ensures fine-grained modality alignment, eliminates data contamination, and provides high usability. Additionally, we develop MM-TSFlib, the first-cut multimodal time-series forecasting (TSF) library, seamlessly pipelining multimodal TSF evaluations based on Time-MMD for in-depth analyses. Extensive experiments conducted on Time-MMD through MM-TSFlib demonstrate significant performance enhancements by extending unimodal TSF to multimodality, evidenced by over 15% mean squared error reduction in general, and up to 40% in domains with rich textual data. More importantly, our datasets and library revolutionize broader applications, impacts, research topics to advance TSA. The dataset is available at `https://github.com/AdityaLab/Time-MMD`.

## 1 Introduction

Time series (TS) data are ubiquitous across a wide range of domains, including economics, urban computing, and epidemiology (52; 56; 29). Analytical tasks on such datasets hence find broad applications in various real-world scenarios such as energy forecasting, traffic planning, and epidemic policy formulation. Human experts typically complete such Time-Series Analysis (TSA) tasks by integrating multiple modalities of time-series data. For instance, epidemiologists combine numerical data on influenza infections with textual domain knowledge, policies, and reports to predict future epidemiological trends. However, most existing TSA models (59; 42; 78; 44; 81; 32; 80; 48) are unimodal, solely using numerical series.

Recently, with the development of Large Language Models (LLMs), the field of TSA is also undergoing an exciting transformative moment with the integration of natural language (73; 34). Existing LLM-based TSA methods incorporate endogenous text derived from numerical series, such as linguistic descriptions of statistical information, which has demonstrated promising benefits (22; 25; 6; 40). However, the potential of exogenous or auxiliary textual signals—such as information on concurrent events and policies that provide additional context to time series—remains untapped. This observation prompts a crucial question for multimodal TSA: **Can multimodal TSA models utilize these**

---

[*]Correspondence to: Haoxin Liu <hliu763@gatech.edu>, B. Aditya Prakash <badityap@cc.gatech.edu>

38th Conference on Neural Information Processing Systems (NeurIPS 2024) Track on Datasets and Benchmarks.

**exogenous textual signals effectively, thereby enhancing current TSA tasks and enabling new applications?**

The primary obstacle in addressing this question lies in the absence of a comprehensive, high-quality multimodal TS dataset, as evidenced by three significant gaps: (1) **Narrow data domains.** Data characteristics and patterns vary between different domains, such as the periodicity of numerical data and the sparsity of textual data. However, current multimodal TS datasets (15; 70; 11; 49; 6) focus solely on stock prediction tasks in the financial domain, which are unable to represent the diverse data domains. (2) **Coarse-grained modality alignment.** Existing multimodal TS datasets only ensure that the text and numerical data come from the same domain, such as general stock news and the prices of one specific stock. Clearly, an abundance of irrelevant text diminishes the effectiveness of multimodal TSA. (3) **Inherent data contamination.** Existing multimodal TS datasets overlook two main reasons of data contamination: (1) Textual data often contains predictions. For example, influenza outlook is a regular section in influenza reports. (2) Outdated test set, particularly the textual data, may have been exposed to LLMs, which are pretrained on vast corpuses. For example, the knowledge cutoff for Llama3-70B is December 2023, which is later than the cutoff dates for most existing multimodal TS datasets. These reasons lead to biased evaluations of general or LLM-based TSA models.

To address the identified gaps, this work aims to introduce a comprehensive, high-quality multimodal TS dataset that spans diverse domains and can be validated through its effectiveness and benefits for TSA. The main contributions of our work are:

- **Pioneering Multi-Domain Multimodal Time-Series Dataset.** We introduce Time-MMD, the first multi-domain multimodal time-series dataset that addresses the aforementioned gaps: (1) encompasses 9 primary data domains. (2) ensures fine-grained modality alignment through meticulously selected data sources and rigorous filtering steps. (3) disentangles facts and predictions from text; ensures all cutoff dates are up to May 2024. To the best of our knowledge, Time-MMD stands as the inaugural high-quality and comprehensive multimodal time-series dataset. We envision Time-MMD offering exciting opportunities to significantly advance time series analysis through multimodal extensions.

- **Pilot Multimodal Time-Series Forecasting Study.** We develop the first-cut multimodal time-series forecasting (TSF) library, MM-TSFlib, piloting multimodal TSA research based on Time-MMD. Our library MM-TSFlib features an end-to-end pipeline with a seamless interface that allows the integration of any open-source language models with arbitrary TSF models, thereby enabling multimodal TSF tasks. MM-TSFlib facilitates easy exploration of Time-MMD and supports future advancements in multimodal TSA.

- **Extensive Evaluations with Significant Improvement.** We conducted over 1,000 experiments of multimodal TSF on Time-MMD using MM-TSFlib. The multimodal versions outperformed corresponding unimodal versions in 95% of cases, reducing the mean squared error by an average of over 20% and up to 40% in some domains with rich textual data. This significant and consistent improvement demonstrates the high quality of Time-MMD, the effectiveness of MM-TSFlib, and the superiority of multimodal extensions for TSF.

We include additional related works in Appendix A and limitations in Appendix B.

## 2 Multi-Domain Multimodal Time-Series Dataset: Time-MMD

We first introduce the key challenges in constructing Time-MMD, followed by the construction pipeline. We then detail each component of the pipeline with corresponding data quality verification. Finally, we discuss considerations for fairness and data release.

**Challenges.** Creating a high-quality, multi-domain numerical-text series dataset presents significant challenges, encompassing the effective gathering, filtering, and alignment of useful textual data. First, textual sources are sparse. Unlike numerical data, typically provided by a "packaged" source, textual data are collected from a variety of dispersed sources, such as reports and news articles, necessitating extensive individual collection efforts. Second, textual information is noisy. Raw textual data often contains large portions of irrelevant information and potential data contamination, such as expert predictions in reports, requiring rigorous filtering processes to ensure data quality. Third, textual data requires precise alignment. It is essential to achieve temporal alignment between textual and numerical data by synchronizing reported times with numerical time steps (e.g., the time

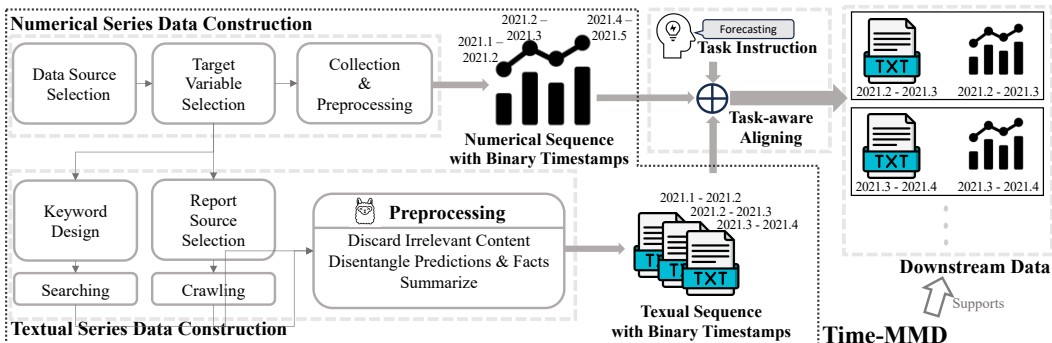

Figure 1: Overview of the Time-MMD construction. We first construct numerical data, then construct textual data from search and report sources with LLM preprocessing targeted at the numerical data, and finally annotate the data with binary timestamps to support various downstream tasks.

step where text is posted) and ensuring that the effective duration of textual information matches the relevant time frames at various granularities (e.g., a seasonal report should correspond to 12 time steps in a weekly time series). Additionally, the dataset faces challenges regarding ease of use, maintenance, and regular updates to remain relevant and useful for ongoing research and applications.

**Pipeline Overview.** We propose a comprehensive pipeline for constructing a text-numeric series dataset utilizing modern LLMs. As illustrated in Figure 1, the construction process is divided into three key steps: (1) Numerical Series Data Construction. We gather numerical data from reputable sources to ensure reliability and accuracy. (2) Textual Series Data Construction. Textual data is collected for fine-grained matching with the numerical data. The quality of this matching is ensured through human selection of data sources and raw text filtering by LLMs. Additionally, LLMs are employed to disentangle facts and predictions and generate summaries. (3) Numerical-Textual Alignment. We use binary timestamps to mark the start and end dates as a universal temporal alignment method between numerical and textual series, supporting the requirements of various downstream TSA tasks.

## 2.1 Numerical Series Data Construction

**Data Source Selection.** We select data sources that are (1) reliable, containing verified knowledge; (2) actively released, allowing for updates with new data; and (3) multi-domain, covering various TSA patterns. Appendix C provides considerations for domain selection. Based on these principles, we choose nine data sources from different domains, as shown in Appendix D. Most sources are from government agencies, with the lowest update frequency being six months.

**Target Variable Selection.** For each domain, we select target variables with significant real-world implications, indicating easier text matching, as shown in Table 1. These variables span three distinct frequencies: daily, weekly, and monthly.

**Collection & Preprocessing.** We collect raw data for all available times, either from batch-released files or through individual scraping. We preprocess the data by discarding early years with a high proportion of missing values. We maintain the original frequency for most domains, adjusting it for security and climate domains due to irregular releases and difficult text matching, respectively. Figure 2 illustrates the diverse patterns present in each domain, such as periodicity and trends.

*Data Quality & Property.* As shown in Table 1 and Figure 2, the constructed numerical data provides comprehensive temporal coverage, ranging from the earliest in 1950 to the present, and exhibits distinct patterns, such as periodicity and trends.

## 2.2 Textual Series Data Construction

**Data Source Selection: Selected Reports and Web Search Results.** The choice of data sources should take into account both extensive coverage and initial strong relevance to the numerical data. Consequently, we combine two appropriate data source types as follows: (1) Selected Reports: For each target variable, we manually select 1-2 highly relevant report series with guaranteed updates. For

Table 1: Overview of numerical data in Time-MMD, covering key variables across nine domains with daily, weekly, or monthly frequencies, sourced from reputable government departments. Eight domains are updated to May 2024; the environment domain update is scheduled for June 2024. We focus on univariate time series forecasting for the target variable, as Time-MMD requires constructing aligned textual data. Covariates are provided for some datasets to inspire future research which are revealed in the dimension column.

| Domain | Target | Dimension | Frequency | Number of Samples | Timespan |
|---|---|---|---|---|---|
| Agriculture | Retail Broiler Composite | 1 | Monthly | 496 | 1983 - Present |
| Climate | Drought Level | 5 | Monthly | 496 | 1983 - Present |
| Economy | International Trade Balance | 3 | Monthly | 423 | 1989 - Present |
| Energy | Gasoline Prices | 9 | Weekly | 1479 | 1996 - Present |
| Environment | Air Quality Index | 4 | Daily | 11102 | 1982 - 2023 |
| Health | Influenza Patients Proportion | 11 | Weekly | 1389 | 1997 - Present |
| Security | Disaster and Emergency Grants | 1 | Monthly | 297 | 1999 - Present |
| Social Good | Unemployment Rate | 1 | Monthly | 900 | 1950 - Present |
| Traffic | Travel Volume | 1 | Monthly | 531 | 1980 - Present |

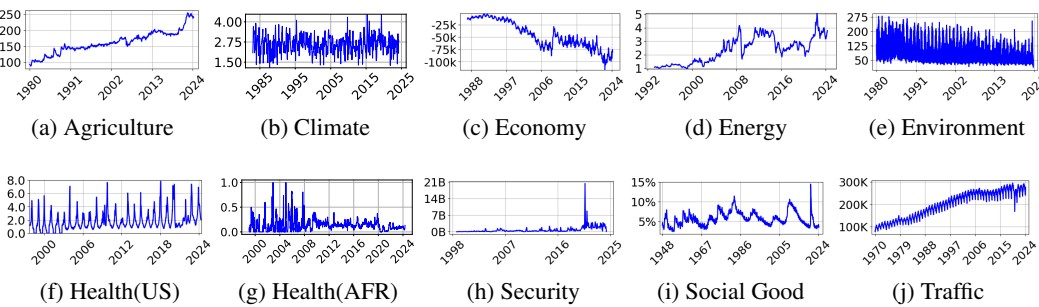

(a) Agriculture    (b) Climate    (c) Economy    (d) Energy    (e) Environment

(f) Health(US)    (g) Health(AFR)    (h) Security    (i) Social Good    (j) Traffic

Figure 2: Visualization of Time-MMD, highlighting distinct characteristics across different domains.

instance, the weekly influenza report[2] published by the Centers for Disease Control and Prevention of the United States is chosen as one of the report sources for the weekly influenza patients proportion of the United States. (2) Web Search Results: For each target variable, we design 2-3 highly relevant keywords used for web searching.

These two data sources complement each other: report data ensures higher relevance but cannot guarantee all-time coverage, while search results cover all times but are highly redundant; search results aggregate multiple data sources, while report data, usually in PDF or TXT format, cannot be extracted by searching. For frequency alignment, the textual data covers multiple frequencies, with search texts enabling daily precision and reports ranging from weekly to monthly and yearly intervals. Appendix D provides a complete list of keywords and reports source.

**Data Collection: Searching and Crawling.** For keyword web searching, we use the official Google API[3] as the entry point. For each keyword, we collect the timestamp, source, title, and content from the top 10 results located each week from 1980 to present. For report data, we parse all available reports from each data source and preserve only plain-text paragraphs.

**Data Preprocessing: Filtering, Disentangling, and Summarizing.** To curate the collected raw text data, we introduce three key preprocessing steps: (1) Filtering to improve relevance; (2) Disentangling facts with predictions to mitigate data contamination; (3) Summarizing for better usability. Given the impracticality of performing these steps manually, we leverage the state-of-the-art LLM, Llama3-70B, to accomplish these tasks.

The prompt used for LLMs is detailed in Appendix F. We incorporate three specific strategies to alleviate the hallucination issue in LLMs and enhance preprocessing quality: (1) A concise introduction of the text. (2) Mandating the LLM to reference the data source, aiding constraint and

---

[2]https://www.cdc.gov/flu/weekly/weeklyarchives2023-2024/week04.htm
[3]https://developers.google.com/custom-search/v1/overview

Table 2: Statistics for text data. Relevance indicates the percentage of text data with relevant content. Coverage describes the proportion of numerical series data being covered by at least one fact. Details are provided in Appendix E. The statistics highlight the need for both reports and search data.

| Source | Raw Tokens | Preprocessed Extracted Facts | | | Preprocessed Extracted Prediction | | |
|--------|-----------|-------------|--------------|--------|-------------|--------------|--------|
| | | Relevance(%) | Coverage (%) | Tokens | Relevance(%) | Coverage (%) | Tokens |
| Report | 17.4k | $84.3_{\pm 27.2}$ | $34.1_{\pm 26.8}$ | $37.6_{\pm 11.9}$ | $82.3_{\pm 28.5}$ | $33.8_{\pm 27.3}$ | $74.6_{\pm 16.1}$ |
| Search | 54.4 | $16.8_{\pm 3.3}$ | $90.7_{\pm 13.5}$ | $38.4_{\pm 4.0}$ | $16.0_{\pm 4.3}$ | $81.9_{\pm 17.2}$ | $62.8_{\pm 3.3}$ |

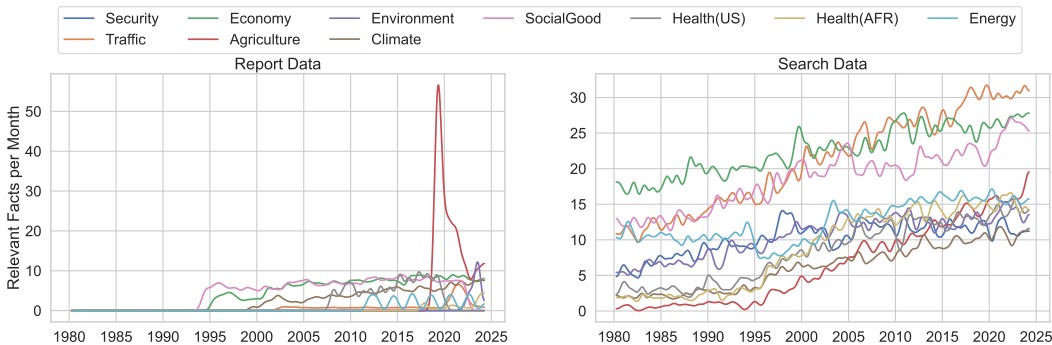

Figure 3: Visualization of relevant report (a, left) and search (b, right) counts in Time-MMD over time. Text counts from both reports and searches increase over time. Domains receiving more attention, such as the economy, contain more available relevant text data..

verification. (3) Permitting the LLM to indicate 'not available' when relevance is uncertain, to avoid fabrication. Appendix G provides a showcase of the text before and after processing.

*Data Quality & Property.* Overall, Figure 3 visualizes the extracted fact count per month over time by domain. Note that the Agriculture report data is of high volume around 2020 and therefore produces a peak. We make the following observations: (a) The search data count exhibits a gradual increasing trend, benefiting from the development of the Internet; the report data count has stabilized in recent years, indicating that release schedule has become stable. (b) The sparsity of textual data varies across different domains, with high-profile fields often accompanied by richer textual data. These validate the extensibility and updatability of Time-MMD and highlight the importance of its coverage across 9 diverse domains.

We further validate the effectiveness of key steps in textual data construction:

(a) Data sources selection. We use *relevance* and *coverage* ratio to describe the percentage of relevant texts and the proportion of numerical series data being covered by at least one fact, respectively. As demonstrated by Table 2, report data exhibits higher relevance but lower coverage; search data display the opposite pattern. Thus, our combined usage serves as a comprehensive solution.

(b) Data preprocessing. Figure 4 provides word cloud visualizations of constructed text data in the health domain, respectively for extracted facts, extracted predictions, and discarded text. Recall that the target variable here is the influenza patients proportion. Highly relevant words such as "pandemic", "vaccine", and "flu" appear more frequently in the extracted facts; research paper-related words such as "edu", "mdpi", and "university" are more common in the discarded text. Besides, the prediction text primarily contains words describing future, such as "will" and "next". These validate the effectiveness of LLM filtering and disentangling. Furthermore, Table 2 presents a comparison of the token count before and after preprocessing. The substantial decrease validates that LLM summarizing improves usability. Appendix H provides the manual verification results on a subset of the data to further validate the effectiveness of preprocessing using LLMs.

### 2.3 Binary Time Stamps for Diverse TSA Tasks

To enable the Time-MMD for versatile and flexible use, we maintain binary timestamps for all numerical and text data, storing the manually verified start dates and end dates. Such binary stamps can be easily referred to while serving different tasks. For report text data, we manually verify the

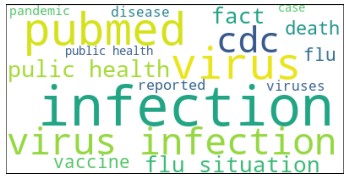

(a) Word cloud visualization for extracted facts

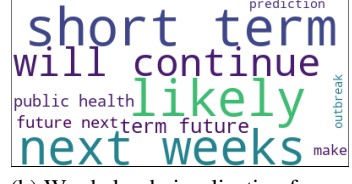

(b) Word cloud visualization for extracted predictions

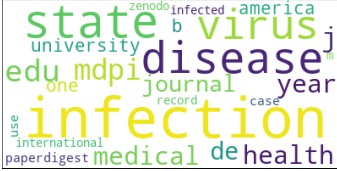

(c) Word cloud visualization for extracted discarded text

Figure 4: Word cloud visualization for influenza patient proportion from the health domain. The discarded texts are identified by the LLM as irrelevant to the target variable. The results validate the effectiveness of LLM preprocessing.

timestamps based on the release notes or report contents. For search data, we integrate adjacent search results within each week and mark timestamps correspondingly.

### 2.4  Considerations for Fairness and Data Release

To consider fairness, we gather data from both the United States and African regions in the Health domain. As depicted in Figure 2, the numerical data of African region exhibits weaker periodicity. Figure 3 shows that the African region has considerably fewer reports compared to the United States. We encourage researchers to consider underrepresented groups when conducting multimodal TSA tasks.

To support various existing and potential novel TSA tasks, we include the following metadata when releasing Time-MMD: (1) Numerical Data: start & end time, target variable, other variables; (2) Text Data: start & end time, fact text (content & data source), prediction text (content & data source).

## 3  Multimodal Time-Series Forecasting Library: MM-TSFlib

In this section, we aim to illustrate the potential benefits of our Time-MMD for multimodal TSA by focusing on time-series forecasting (TSF), a fundamental TSA task. TSF involves predicting future events or trends based on historical time-series data. While most existing TSF methods primarily depend on numerical series, we aim to extend these unimodal TSF methods to multimodality. To achieve this, we contribute both formulating the multimodal TSF problem as well as introducing MM-TSFlib, the first-cut reasonable approach for multimodal TSF.

### 3.1  Problem Formulation

Conventional unimodal TSF models take a numerical series as input and output future values of some or all of its features. Let the input variable of the numerical series be denoted as $\boldsymbol{X} \in \mathbb{R}^{l \times d_{\text{in}}}$, where $l$ is the length of the *lookback window* decided by domain experts and $d_{\text{in}}$ is the feature dimension at each time step. The output variable of the forecasts generated of *horizon window* length $h$ is denoted as $\boldsymbol{Y} \in \mathbb{R}^{h \times d_{\text{out}}}$, where $d_{\text{out}}$ is the dimension of targets at each time step. For the sample at time step $t$, denoted as $(\mathbf{X}_t, \mathbf{Y}_t)$, $\mathbf{X}_t \in \boldsymbol{X} = [\mathbf{x}_{t-l+1}, \mathbf{x}_{t-l+2}, \ldots, \mathbf{x}_t]$ and $\mathbf{Y}_t \in \boldsymbol{Y} = [\mathbf{y}_{t+1}, \mathbf{y}_{t+2}, \ldots, \mathbf{y}_{t+h}]$. Thus, the unimodal TSF model parameterized by $\theta$ is denoted as $f_\theta : \mathcal{X} \to \mathcal{Y}$.

For multimodal TSF, the input variable of the textual series is also considered, which can be denoted as $\boldsymbol{S} \in \mathbb{R}^{k \times d_{\text{txt}}}$, where $k$ is the lookback window length of the text series, independent of $l$, and $d_{\text{txt}}$ is the feature dimension of the text. Although the text variable may have inconsistent feature dimensions, we slightly abuse the notation $d_{\text{txt}}$ here for brevity. Thus, the multimodal TSF model parameterized by $\theta$ is denoted as $g_\theta : \mathcal{X} \times \mathcal{S} \to \mathcal{Y}$.

### 3.2  Pioneering Solution for Multimodal TSF

**Multimodal Integration Framework.** We propose a pioneering multimodal integration framework to extend existing unimodal TSF models to their multimodal versions. As illustrated in Figure 5, our framework features an end-to-end pipeline that integrates open-source language models with

various TSF models. Numerical and textual series are independently modeled using unimodal TSF models and LLMs with projection layers. These outputs are then combined using a learnable linear weighting mechanism to produce the final prediction. To reduce computational costs, we freeze the LLM parameters and train only the additional projection layers. We employ pooling layers to address the inconsistent dimensions of textual variables. This framework features an end-to-end training manner, with minimal overhead in trainable parameters.

**Multimodal TSF Library**. Building upon the multimodal dataset Time-MMD and integration framework, we present the first multimodal TSF library, named **MM-TSFlib**. MM-TSFlib supports multimodal extensions of over 20 unimodal TSF algorithms through 7 open-source (large) language models, including BERT (14), GPT-2 (50) (Small, Medium, Large, Extra-Large), Llama-2-7B (58), and Llama-3-8B[4]. We detail the implementations and language models in Appendix J.

MM-TSFlib is designed for ease of use with Time-MMD in multimodal TSA. Additionally, MM-TSFlib serves as a pilot toolkit for evaluating the multimodal extensibility of existing TSF models.

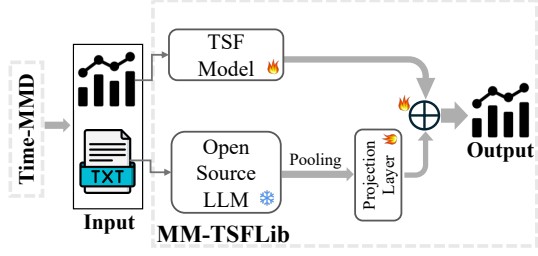

Figure 5: Overall structure of the MM-TSFlib. MM-TSFlib uses a model-agnostic multimodal integration framework that independently models numerical and textual series within an end-to-end training manner. MM-TSFlib slightly increases the number of trainable parameters, balancing effectiveness and efficiency.

Figure 6: Average normalized MSE results for each TSF backbone. Blue areas represent the performance gap between unimodal and multimodal results. The multimodal experiments significantly and consistently outperform corresponding unimodal ones. Detailed results are provided in Appendix O

# 4 Experiments for Multimodal TSF

Based on the constructed MM-TSFlib, we further conduct comprehensive experiments to demonstrate the superiority of multimodal TSF and the high quality of Time-MMD."We further investigate the impact of data domains, horizon window size and the text modeling method.

## 4.1 Experimental Setup

We adhere to the general setups following existing TSF literatures (67; 66; 48). Regarding the horizon window length, we consider a wider range **from short- to long-term TSF** tasks, with four different lengths for each dataset according to frequency. We conduct TSF tasks on **all 9 domains of Time-MMD**. We employ the widely-adopted mean squared error (MSE) as the evaluation metric. A higher MSE indicates a better performance.

We comprehensively consider **12 advanced unimodal TSF methods across 4 types** including: (1) Transformer-based: Transformer (59), Reformer (32), Informer (80), Autoformer (67), Crossformer (78), Non-stationary Transformer (44),FEDformer (82), iTransformer (42). (2) MLP-based: DLinear (74). (3) Agnostic: FiLM (81). (4) LLM-based: Time-LLM (25). Unless otherwise specified, we use GPT-2-Small as the LLM backbone in MM-TSFlib. We deployed two sets of experiments upon each TSF model, i.e., both unimodal and multimodal versions. More details about experimental setup are provided in Appendix K.

---

[4]https://llama.meta.com/llama3

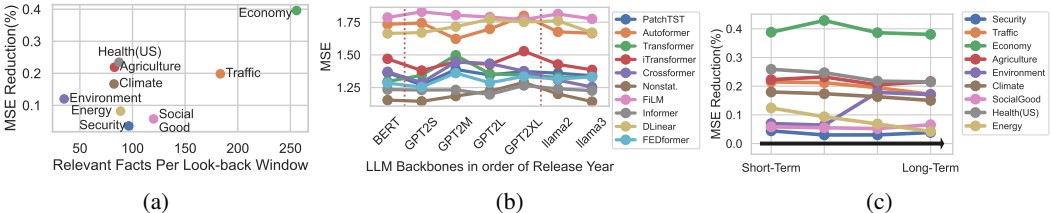

Figure 7: Results of exploratory experiments. (a) Influence of data domains: inherent characteristics and text richness influence performance. (b) Influence of LLM backbones: unclear correlation between multimodal TSF performance and LLM natural language capabilities. (c) Influence of horizon window size: improvements via multimodality are robust to horizon window size.

## 4.2 Experimental Results

Our experiments aim to investigate the following five aspects. Appendix O provides detailed results.

**Effectiveness of multimodal TSF.** Figure 6 shows average MSE results for corresponding unimodal and multimodal versions of each TSF backbone. The multimodal versions consistently outperform corresponding unimodal versions. As detailed in Appendix O, the multimodal versions outperformed their unimodal versions in **95%** of over 1,000 experiments, reducing the mean squared error by over **15%** in average and up to **40%** in domains with rich textual data. Such consistent improvements fully validate the superiority of multimodal TSF and the effectiveness of our proposed multimodal framework in Section 3.

Furthermore, we observe that different TSF backbones benefit from multimodal extension to varying degrees. For example, the originally inferior Informer exhibits strong multimodal performance, which we attribute to its intrinsic design for modeling long-range dependencies that may benefit more from textual cues.

Additional experiments on text integration approach approaches are provided in Appendix L. We hope these results inspire more advanced multimodal TSF solutions.

**Quality of Time-MMD dataset.** Figure 6 shows that SOTA unimodal TSF models, such as iTransformer and PatchTST, maintain leading unimodal performance, validating the quality of Time-MMD's numerical data. Moreover, multimodal extension consistently and significantly improves performance by incorporating textual data, confirming the quality of Time-MMD's textual data.

**Influence of data domains.** Figure 7a shows the relationship between the relevant fact count and the reduced MSE via multimodal extension for each domain. The scatter plot generally illustrates a positive linear correlation, aligning with the innovation of integrating textual information. Besides, domain characteristics also influence multimodal performance, even with a similar fact count. For example, the security domain, focusing on disasters and emergency grants, exhibits higher unpredictability in the future thus benefits less from the historical textual information. This observation highlights the importance of Time-MMD's coverage of 9 diverse domains.

**Influence of the horizon window size.** Figure 7c shows the relationship between horizon window size and the average MSE reduction for each domain. Overall, the MSE reduction is stable and promising across different horizon window size, from short term to long term. These results demonstrate that the effectiveness of multimodal TSF is robust to different forecast horizon requirements.

**Influence of the text modeling method.** Firstly, we varied the LLM backbone in MM-TSFlib and evaluated corresponding multimodal performance on health domain. As shown in Figure 7b, the choice of LLM backbone does not exhibit a significant correlation with multimodal TSF performance. For the GPT2 series, the scaling law is unclear for multimodal TSF, indicating no clear positive correlation between the parameter scale and TSF performance. Across different LLMs, multimodal TSF performance is relatively similar, even between the advanced Llama-3-8B and the earlier BERT. There might be three possible reasons: (1) Our proposed multimodal framework, although effective, still does not fully utilize the power of LLMs, particularly by only fine-tuning through projection layers. (2) Existing LLMs, pre-trained for natural language tasks, may not be directly suitable for multimodal TSF. (3) The embedding dimension of BERT is 768, much lower than the 4096 dimension

size of Llama-3-8B, thereby makes it easier to fine-tune an effective projection layer with limited training data.

Although LLMs are currently the mainstream approach for text modeling, they may not be suitable for certain scenarios, such as unfamiliar domains or unsupported languages. Therefore, we introduce Doc2Vec (33) a text embedding model trained from scratch, as an alternative for text modeling. The experimental results on three datasets across four prediction horizons are provided in Table 6. The results show that Doc2Vec is also effective but generally performs worse than BERT. Doc2Vec is further included in our MM-TSFlib, which will greatly enhance applicability to underrepresented languages and domains.

We provide additional experiments and discussions on multimodal modeling approaches in Appendix N, including the introduction of attention mechanisms (59) and the use of closed-source LLMs, such as GPT-3.5[5].

In summary, all these observations suggest that there is significant room for improvement in the methodological research for multimodal TSF.

# 5 Potential Future Works

Beyond its efficacy in enhancing time-series forecasting accuracy through multimodality extension (Section 4), Time-MMD holds significant potential in advancing time series analysis across a wide spectrum. In the following section, we discuss how Time-MMD might transform conventional approaches, facilitate novel methodologies, and may broadly impact the time series analysis domain in future works.

**Multimodal Time-Series Imputation.** Missing values in time series data, caused by sensor failures, system instability, or privacy concerns, pose a significant challenge in analysis. Conventional time-series imputation (TSI) methods (8; 19; 3; 4) often overlook valuable information captured in textual formats alongside the numerical data. For instance, incident reports, weather conditions, and special events can provide crucial context for imputing missing data points in traffic time series, but current methods fail to effectively incorporate this information. By constructing missing values using existing toolkits (16), Time-MMD can directly serve as multimodal imputation datasets. Time-MMD enables the integration of textual contextual information with numerical time series data, opening new avenues for multimodal time series analysis and enhancing imputation accuracy.

**Multimodal Time-Series Anomaly Detection.** Detecting anomalies in time series data is crucial for identifying unusual patterns that may indicate faults, fraud, or other significant events (10; 5; 75). However, conventional anomaly detection methods (47; 7; 69; 53) are limited to expected pattern deviations in numerical data, overlooking valuable information in textual formats. For instance, news articles, social media posts, and market reports can provide critical context that influences financial market behavior and helps identify anomalies not evident from numerical data alone. Time-MMD enables a feasible way to support multimodal time series anomaly detection. For example, in influenza (health) dataset, if "flu outbreak" is mentioned in the text data, we can initially label the aligned numerical data as anomalous.

**Multimodal Foundation Time-Series Models.** The introduction of Time-MMD, a comprehensive text-numeric TS dataset, is expected to significantly advance multimodal-based TS methods, including the development of multimodal foundation TS models beyond existing unimodal models (30). Time-MMD will facilitate further exploration with more complex and informative prompts, enhancing the performance and capabilities of fine-tuning methods. Additionally, it might spur research and development of multimodal models specifically tailored for TSA, an area with limited exploration compared to other domains like vision and video generation (54; 62; 68; 17).

# 6 Ethics Statement

While collecting data from government and news websites, we rigorously adhered to ethical standards to ensure compliance with website policies and avoid potential conflicts of interest. Mindful of copyright and regional policies, we restricted our collection to content freely available without

---

[5]https://platform.openai.com/docs/models/gpt-3-5-turbo

premium access or subscription requirements. In collecting data from web searches, we used Google's official API to ensure that the data strictly complied with ethical standards.

## 7 Conclusion

In this work, we propose Time-MMD, the first multi-domain multimodal time series dataset, and develop MM-TSFlib, the first multimodal time series forecasting library, which facilitates a pilot study for multimodal time series analysis on Time-MMD. We conduct extensive experiments to demonstrate the high quality of Time-MMD, the effectiveness of MM-TSFlib, and the superiority of integrating textual information for time series analysis. We envision that this work catalyzes the transformation of time series analysis from unimodal to multimodal by integrating natural language.

**Acknowledgements:** This paper was supported in part by the NSF (Expeditions CCF-1918770, CA-REER IIS-2028586, CAREER IIS-2144338, Medium IIS-1955883, Medium IIS-2106961, Medium IIS-2403240, PIPP CCF-2200269), CDC MInD program, Meta faculty gifts, and funds/computing resources from Georgia Tech and GTRI. This work used the Delta GPU Supercomputer at NCSA of UIUC through allocation CIS240288 from the Advanced Cyberinfrastructure Coordination Ecosystem: Services & Support (ACCESS) program.

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

## Appendix

## A  Additional Related Dataset Work

Existing datasets (2; 71; 51; 18; 55; 15; 6) primarily focus on stock analysis tasks in the finance domain. Other multi-modal datasets for traffic demand prediction (51), news impact prediction (24) and electrocardiogram classification (34) are proposed recently. However, these datasets still do not address the aforementioned gaps. Especially constructing datasets for diverse domains, such as agriculture and security, is more challenging but holds substantial real-world impact.

Besides, some datasets focus on combining images with numerical sequences, where the images are typically sourced from spatial domains, such as satellite imagery. These works (21; 37) are centered on spatiotemporal data analysis, whereas our work focuses on time series analysis.

Additional multi-modal datasets (24) for news impact prediction and electrocardiogram classification (34) are proposed recently. However, these datasets still do not address the aforementioned gaps. Especially constructing multimodal datasets for other domains, such as agriculture and security, is more challenging but holds substantial real-world impact.

## B  Limitations

Our work provides a comprehensive, high-quality multimodal time series dataset, but it still has limitations in terms of dataset diversity, as all the text data comes from English. We plan to extend Time-MMD to multilingual versions to better address diversity and leverage data from multiple languages (26). In addition, utilizing Time-MMD to support other multimodal time series analysis tasks, such as multivariate time series forecasting and time series classification, requires additional dataset curation work. Moreover, considering multimodal time series datasets with images and audio is a worthwhile direction for future work. In this work, we chose to include text for the following reasons: (1) Text and numerical series commonly coexist across multiple domains. (2) With the rise of LLMs, an increasing number of time-series methods are incorporating text based on large language models. Collecting data that further align images (21) ,audio (31) or graph (39) with text and numeric is meaningful but beyond the scope of this work. Lastly, this work focuses on time series forecasting and does not cover interacting dynamical systems or equation discovery (46), where language could provide insights into the qualitative behavior of the system.

Our multimodal time series forecasting library is built upon a simple integration framework that only uses a projection layer for fine-tuning LLMs. How to fine-tune LLMs more efficiently and effectively for time series analysis remains an interesting topic. Moreover, addressing the issue of distributional shift (77; 79; 38) in the text should also be considered in future work to build robust models. In summary, our approach to achieve multimodal time series forecasting based on Time-MMD dataset is a first-cut reasonable approach rather than the optimal solution. There could be better ways to utilize it, such as Retrieval-Augmented Generation (20).

Despite these limitations, we hope that the datasets and library we have constructed will facilitate broader research and applications in multimodal time series analysis, such as benchmark (27) and agent (28).

## C  Considerations for Domain Selection

Our domain selection is based on the following three considerations:

- Most of our selected domains are also widely used in existing time-series works (45; 1; 13; 65) but lack textual and numerical alignment. We chose these domains to facilitate comparison by researchers, and all these domains have significant real-world importance.
- The remaining three domains, including social good, agriculture, and security, are often overlooked by researchers. We included these domains to contribute to a more comprehensive dataset.
- Additionally, we also considered underrepresented groups, as reflected in the Health (Africa) dataset, compared with the Health (U.S.).

These domains exhibit differences in numerical data, textual data, and performances. To the best of our knowledge, our Time-MMD is the most domain-inclusive multimodal time-series dataset.

# D    Details of Data Sources and Variables by Domain

## D.1    Agricultural Domain Dataset Sources

### D.1.1    Numeric Data

The sequence raw data is sourced from the *retail broiler composite*, provided by the United States Department of Agriculture (USDA) Economic Research Service (ERS). The raw data can be accessed at the USDA ERS website[6].

### D.1.2    Text Data

**Report: Source 1**

- **Frequency:** Weekly
- **Name:** USDA Broiler Market News Report.
- **URL:** `https://usda.library.cornell.edu/concern/publications/5999n3392?locale=en&page=30#release-items`.
- **Description:** This report provides a summary of the national broiler market, with data consolidated from a number of linked reports. The release contains price information for broiler parts by U.S. region as well as in Mexico City, slaughter estimates, availability, and national fowl market and miscellaneous poultry information.

**Report: Source 2**

- **Frequency:** Daily
- **Name:** Daily National Broiler Market at a Glance.
- **URL:** `https://usda.library.cornell.edu/concern/publications/vx021f146?locale=en&page=142#release-items`.
- **Description:** This report summarizes the daily trends in the national broiler market, with commentary on price, demand and supply, market activity, and trade.

**Search**

- **Frequency:** 10 results per week.
- **Keywords:** United States Broiler Market; Chicken prices

## D.2    Climate Domain Dataset Sources

The data is collected by the NOAA National Centers for Environmental Information and spans from 1895 to the present. It offers a comprehensive historical dataset for analyzing long-term climate trends and their implications for energy and environmental policies. The raw data can be accessed at the NOAA.

### D.2.1    Numeric Data

**Source**  :`https://www.drought.gov/historical-information?dataset=0&selectedDateUSDM=20240514`

---

[6]`https://www.ers.usda.gov/data-products/meat-price-spreads/documentation/`

### D.2.2 Text Data

**Report: Source 1**

- **Frequency:** Month
- **Name:** Drought Report of NOAA National Centers for Environmental Information[7].
- **Description:** Drought Highlights;National Overview;Regional Overview

**Report: Source 2**

- **Frequency:** Month
- **Name:** National Climate Report of NOAA National Centers for Environmental Information[8].
- **Description:** National Overview;Monthly and Seasonal Highlights;

**Search**

- **Frequency:** 10 results per week.
- **Keywords:** Drought and extreme weather; Precipitation

## D.3 Economy Domain Dataset Sources

### D.3.1 Numeric Data

**Source** :`https://www.census.gov/foreign-trade/balance/c0015.html`.

### D.3.2 Text Data

**Report: Source 1**

- **Frequency:** Monthly
- **Name:** U.S. International Trade in Goods and Services [9].

**Report: Source 2**

- **Frequency:** Monthly
- **Name:** Advance Economic Indicators Report[10].

**Search**

- **Frequency:** 10 results per week.
- **Keywords:** International Trade Balance

## D.4 Energy Domain Dataset Sources

This subsection outlines the sources and characteristics of the sequence and raw textual data that form the energy component of the series-text dataset. This domain specifically focuses on gasoline prices, which are crucial for analyzing economic stability, consumer spending patterns, and the overall energy market dynamics.

### D.4.1 Numeric Data

The sequence raw data comprises weekly statistics of U.S. gasoline prices, measured in dollars per gallon. Monitoring these prices provides insights into energy market fluctuations and helps predict changes in economic conditions and policy adjustments.

---

[7]`https://www.ncei.noaa.gov/access/monitoring/monthly-report/drought/200602`
[8]`https://www.ncei.noaa.gov/access/monitoring/monthly-report/national/200602`
[9]`https://www.census.gov/foreign-trade/Press-Release/ft900_index.html`
[10]`https://www.census.gov/econ/indicators/current/index.html`

**Source and Accessibility**  This data is collected by the U.S. Energy Information Administration (EIA) and spans from April 5, 1993, to the present, offering a comprehensive historical view of fuel economics. The raw data can be accessed at the EIA website[11].

### D.4.2  Text Data

We collect raw text from various EIA publications and reports, as they provide contextual information and expert analysis related to the Numeric Data.

**Report Source 1: Annual Energy Outlook**

- **Frequency:** Annually
- **Source:** U.S. Energy Information Administration[12].
- **Type:** Annual Energy Outlook reports
- **Coverage:** From 1979 to present

**Report Source 2: Weekly Petroleum Status Report**

- **Frequency:** Weekly
- **Source:** U.S. Energy Information Administration[13].
- **Type:** Weekly Petroleum Status Reports
- **Coverage:** From 2011 to present

**Search**

- **Frequency:** 10 results per week.
- **Keywords:** Gasoline prices

### D.5  Social Good Domain Dataset Sources

This subsection describes the sources and characteristics of the sequence and raw textual data that comprise the social good component of the series-text dataset. This domain focuses on unemployment statistics in the United States, segmented by various racial groups, reflecting the societal impact of economic disparities.

### D.5.1  Numeric Data

The sequence raw data consists of monthly unemployment statistics for the United States, disaggregated by race. These statistics include data for the following racial groups: White, Black or African American, Asian, American Indian or Alaska Native, and Native Hawaiian or Other Pacific Islander.

This data is sourced from the United States Bureau of Labor Statistics (BLS) and spans from 1954 to the present, providing a long-term view of employment trends across different racial demographics. The raw data can be accessed at the BLS website[14].

### D.5.2  Text Data

The raw textual data is sourced from official reports which provide a comprehensive analysis of employment conditions, enriched with context that is crucial for understanding the nuances of unemployment across different racial groups.

---

[11]https://www.eia.gov/petroleum/gasdiesel/
[12]https://www.eia.gov/outlooks/aeo/archive.php
[13]https://www.eia.gov/petroleum/supply/weekly/archive/
[14]https://data.bls.gov/home.htm

**Report Source 1**

- **Frequency:** Monthly
- **Source:** U.S. Department of Labor, BLS[15].
- **Type:** Official report on monthly employment situations
- **Coverage:** From 1994 to present
- **Quantity:** TBD
- **Relevance:** Directly relevant (Siamese)

**Report Source 2**

- **Frequency:** Annually
- **Source:** U.S. Department of Labor, BLS[16].
- **Type:** Official report on annual labor force characteristics by race and ethnicity
- **Coverage:** From 2015 to present
- **Quantity:** TBD
- **Relevance:** Directly relevant

**Search**

- **Frequency:** 10 results per week.
- **Keywords:** Unemployment rate; Employment situation

### D.6 Public Health (United States) Domain Dataset Sources

This subsection outlines the sources and characteristics of the sequence and raw textual data that form the public health component of the series-text dataset. This domain specifically focuses on Influenza-Like Illness (ILI) statistics, which are crucial for monitoring seasonal and pandemic influenza outbreaks, guiding public health interventions, and planning healthcare resources effectively.

#### D.6.1 Numeric Data

The sequence raw data includes weekly statistics of ILI cases in the United States, sourced from the Centers for Disease Control and Prevention (CDC). ILI, representing a category of health conditions characterized by symptoms similar to those of influenza, serves as an important public health indicator for tracking influenza activity.

**Source and Accessibility**   The data is collected and made available by the CDC's ILINet system, which has been tracking ILI cases since 1954. This extensive dataset allows for robust historical trend analysis and modeling of influenza patterns. The raw influenza patients data can be accessed at the CDC website[17].

#### D.6.2 Text Data

Raw textual data in the public health domain includes official reports and related news, providing insights into the context and implications of the ILI statistics.

**Report Source 1: Weekly U.S. Influenza Surveillance Report**

- **Frequency:** Weekly
- **Source:** CDC[18].

---

[15]https://www.bls.gov/bls/news-release/empsit.htm

[16]https://www.bls.gov/opub/reports/race-and-ethnicity/2015/home.htm

[17]https://gis.cdc.gov/grasp/fluview/fluportaldashboard.html

[18]https://www.cdc.gov/flu/weekly/pastreports.htm

- **Type:** Official Weekly U.S. Influenza Surveillance Report
- **Coverage:** From 1999 to present

## Report Source 2: Annual Flu Season Key Studies and News Reports

- **Frequency:** Annually
- **Source:** CDC[19].
- **Type:** Key studies and news reports from the flu season
- **Coverage:** From 2019 to present

## Search

- **Frequency:** 10 results per week.
- **Keywords:** influenza; epidemic

### D.7 Public Health (Africa) Domain Dataset Sources

#### D.7.1 Numeric Data

**Source:** World Health Organization [20].

#### D.7.2 Text Data

**Report: Source 1**

- **Frequency:** Weekly
- **Name:** Influenza Virological Surveillance in the WHO African Region[21].

**Report: Source 2**

- **Frequency:** Daily
- **Name:** Press of Africa Centres for Diseases Control[22].
- **Description:**

## Search

- **Frequency:** 10 results per week.
- **Keywords:** influenza; epidemic

### D.8 Environment Domain Dataset Sources

#### D.8.1 Numeric Data

**Source** :`https://www.epa.gov/outdoor-air-quality-data`.

#### D.8.2 Text Data

**Report: Source 1**

- **Frequency:** Daily
- **Name:** Press about Air Quality of Department of Environmental Conservation of New York[23].

---

[19]`https://www.cdc.gov/flu/spotlights/index.htm`

[20]`https://app.powerbi.com/view?r=eyJrIjoiNjViM2Y4NjktMjJmMC00Y2NjLWFmOWQtODQ0NjZkNWM1YzNmIiwidCI6ImY2MTBjM`

[21]https://www.afro.who.int/publications/influenza-virological-surveillance-who-african-region-epidemiological-week-39-2017

[22]`https://africacdc.org/news-item/africa-centres-for-diseases-control-and-prevention-launches-new-network`

[23]https://dec.ny.gov/news/press-releases/2021/5/dec-directs-pvs-chemical-solutions-inc-to-temporarily-cease-operations

**Report: Source 2**

- **Frequency:** Daily
- **Name:** Article about air quality from New York National Broadcasting Company[24].

**Search**

- **Frequency:** 10 results per week.
- **Keywords:** New York air quality; New York air pollution

## D.9    Traffic Domain Dataset Sources

### D.9.1    Numeric Data

**Source**   :`https://www.fhwa.dot.gov/policyinformation/travel_monitoring/tvtfaq.cfm`.

### D.9.2    Text Data

**Report: Source 1**

- **Frequency:** Weekly
- **Name:** Weekly Traffic Volume Report[25].
- The Traffic Volume report estimates the vehicle miles traveled (VMT) for interstate highways and how the total travel measured by VMT compares with travel that occurred in the same week of the previous year. The VMT is further split into passenger vehicle and truck components. The information gives new insights into the effect on traffic by storm activity, economic fluctuations, and other variables that could not be evaluated using the monthly report.

**Search**

- **Frequency:** 10 results per week.
- **Keywords:** Travel; Mobility

## D.10    Security Domain Dataset Sources

### D.10.1    Numeric Data

**Source**   :`https://www.fema.gov/about/openfema/data-sets`.

### D.10.2    Text Data

**Report: Source 1**

- **Frequency:** Uncertain
- **Name:** Billion-Dollar Weather and Climate Disasters[26].

**Report: Source 2**

- **Frequency:** Uncertain
- **Name:** Disaster and emergency declarations[27].

---

[24]`https://www.nbcnewyork.com/tag/air-quality/`
[25]https://datahub.transportation.gov/stories/s/Weekly-Traffic-Volume-Report/3g63-ik4u/
[26]https://www.ncei.noaa.gov/access/billions/
[27]`https://www.fema.gov/about/openfema/data-sets`

# E    Statistics of Textual Data

We show statistics of collected report data and search data in Table 3 and Table 4. As the security reports themselves contain manually-written summaries, we didn't perform a LLM preprocessing on them.

Table 3: Detailed relevance ratio, coverage ratio and average token counts of LLM-processed report data on each domain. Token counts are collected by GPT2Tokenizer from Huggingface (63). As the collected security reports already contain hand-written summaries, we didn't perform a LLM preprocessing on them.

| Domain | Raw Tokens | Fact | | | Prediction | | |
|---|---|---|---|---|---|---|---|
| | | Relevance(%) | Coverage(%) | Tokens | Relevance(%) | Coverage(%) | Tokens |
| Agriculture | 3850.62 | 99.81 | 10.08 | 31.63 | 99.67 | 10.08 | 65.71 |
| Climate | 6501.57 | 98.64 | 61.09 | 36.71 | 98.64 | 61.09 | 69.99 |
| Economy | 62315.99 | 100.00 | 81.56 | 31.55 | 100.00 | 81.56 | 66.11 |
| Energy | 2335.59 | 100.00 | 23.94 | 45.97 | 100.00 | 23.94 | 92.53 |
| Environment | 495.24 | 90.38 | 1.80 | 44.75 | 75.00 | 1.53 | 82.33 |
| Health (US) | 3705.83 | 56.25 | 34.63 | 33.68 | 56.37 | 34.70 | 62.10 |
| Health (AFR) | 934.50 | 21.99 | 6.25 | 35.53 | 17.84 | 3.12 | 91.19 |
| Security | 10.78 | - | - | - | - | - | - |
| Social Good | 75878.06 | 93.53 | 37.56 | 18.17 | 93.53 | 37.56 | 47.58 |
| Traffic | 1002.40 | 97.82 | 50.28 | 60.69 | 100.00 | 50.28 | 93.68 |

Table 4: Detailed relevance ratio, coverage ratio and average token counts of LLM-processed search data on each domain

| Domain | Raw Tokens | Fact | | | Prediction | | |
|---|---|---|---|---|---|---|---|
| | | Relevance(%) | Coverage(%) | Tokens | Relevance(%) | Coverage(%) | Tokens |
| Agriculture | 54.26 | 10.58 | 81.85 | 35.39 | 10.96 | 61.09 | 64.76 |
| Climate | 54.33 | 12.49 | 98.59 | 35.55 | 8.17 | 57.06 | 67.91 |
| Economy | 53.16 | 17.25 | 94.09 | 35.97 | 16.86 | 91.25 | 63.05 |
| Energy | 55.45 | 19.12 | 78.43 | 36.64 | 16.85 | 76.81 | 57.62 |
| Environment | 55.61 | 16.20 | 95.97 | 37.48 | 15.92 | 83.22 | 66.60 |
| Health(US) | 56.46 | 19.49 | 99.71 | 48.40 | 21.12 | 97.77 | 60.05 |
| Health(AFR) | 56.66 | 22.20 | 99.39 | 41.99 | 21.48 | 95.20 | 63.95 |
| Security | 51.49 | 16.61 | 99.66 | 37.97 | 12.39 | 97.64 | 64.48 |
| SocialGood | 52.67 | 17.21 | 59.11 | 37.42 | 19.22 | 59.11 | 59.35 |
| Traffic | 53.42 | 17.12 | 100.00 | 36.89 | 17.46 | 99.62 | 60.53 |

## F    Prompt designed for LLM preprocessing

```
Instructions:
You are expert of {domain}.
Instructions:
1. Carefully read through the following {reportname} from {start_date} to {
     end_date} published by {Author}. Description: {Description}. Due to the
     length of the report, only an excerpt is provided below.
2. Filter the results to find information useful to making predictions about
     {keyword}. Discard any irrelevant information.
3. Summarize the useful filtered information into the following 5 parts:
   - Objective facts about the {keyword} situation
   - Analysis of the current situation
   - Predictions for the short-term future (next {short_term})
   - Predictions for the long-term future (next {long_term})
4. Format your output as follows:
   - Start each part with a tag indicating the type of content, using these
       tags:
       - #F# for objective facts
       - #In# for insights
       - #A# for analysis of current situation
       - #SP# for short-term predictions
       - #LP# for long-term predictions
     - Write each part concisely, using no more than 3 sentences.
     - For objective facts, cite the source at the end using a [Source] tag.
     - For your own opinions, use a [LLM] tag.
     - If no useful information can be found, simply write "NA" for that
         part.
Remember, focus only on information relevant to predicting {keyword}. You are
     allowed to use "NA".
```

Figure 8: LLM prompt template used for preprocessing.

We show the prompt we use while doing LLM preprocessing in Figure 8. Corresponding keywords come from manually written domain-specific instructions.

# G    Showcase of Raw and Preprocessed Text

We show the pre and post-processing text content of a report in Climate domain in Figure 9. We observe that LLMs are able to accurately summarized the factual content from original report.

```
Raw Content:

About 45% of the contiguous U.S. fell in the moderate to extreme drought
    categories (based on the Palmer Drought Index) at the end of June.
...
According to the weekly U.S. Drought Monitor (USDM), as of June 28, 2022,
    47.73% of the contiguous U.S. (CONUS) (42.53% of the U.S. including
    Alaska, Hawaii, and Puerto Rico) was classified as experiencing moderate
    to exceptional (D1-D4) drought.
...

LLM Summarization:

...
#F# The national proportion of dry areas was about 45% of the contiguous U.S.
    at the end of June 2022, with 47.73% of the contiguous U.S. experiencing
    moderate to exceptional (D1-D4) drought.
...
```

Figure 9: Show case of a text report before and after LLM preprocessing, sampled from Climate domain.

# H    Manual Verification of LLM Preprocessing

To validate the quality of LLM preprocessing, we manually inspected 100 text samples. Our observations are as follows:

- Among the 127 extracted facts, 8 were fabricated by the LLM.
- In the 8 fabricated instances, 5 were labeled with the data source as LLM, indicating that they can be filtered out. This results in a true hallucination rate of 3/127.
- Among the 52 text samples discarded by the LLM, 4 were manually identified as containing relevant information, yielding an error discard rate of 4/52.

# I   Case Study of Semantic Alignment

A case example of textual data explaining numerical data changes is shown in Figure 10 from the Health dataset. As the reports and search data are collected while ensuring strong time alignments, their textual content can provide clear illustrations on numerical series changes. For example, during Feb 2008, the health care numerical data experienced a significant upward trend, while the summarized report data states "...... The proportion of deaths attributed to pneumonia and influenza was above the epidemic threshold for the fourth consecutive week ......". Explicit alignments like this can be easily found in our data.

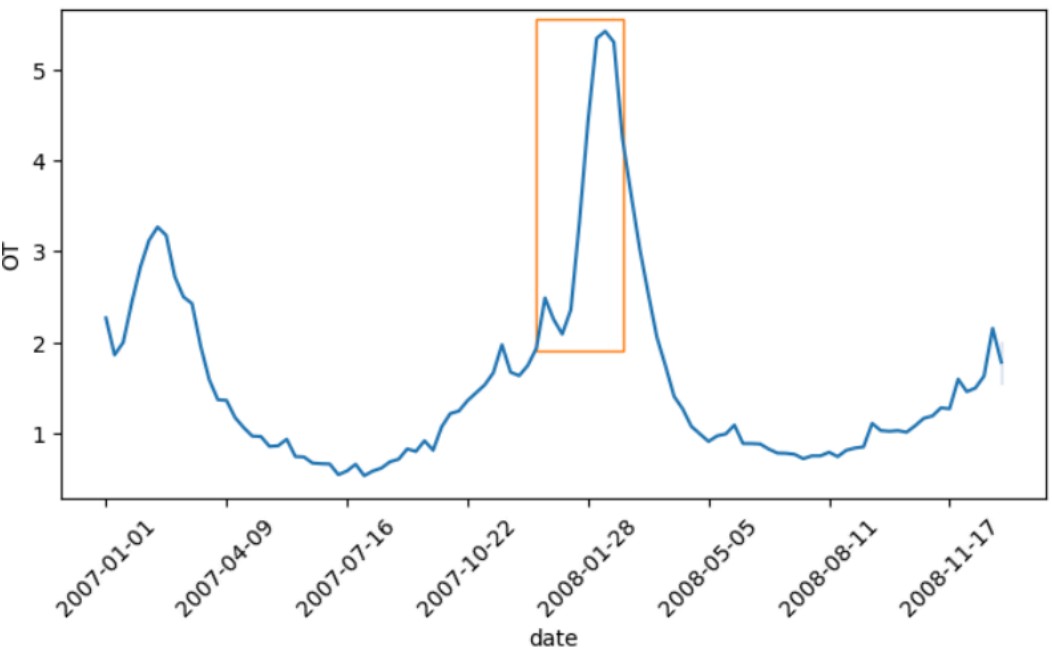

Figure 10: Case Study of Semantic Alignment from the Health Dataset.

# J   Details of MM-TSFlib

In terms of implementation, MM-TSFlib chooses to extend the widely used Time-Series Library (TSlib)[28], thus ensuring ease of use. For LLM invocation, MM-TSFlib utilizes the popular and active Hugging Face[29]. For the projection layer, we use a multilayer perceptron (MLP) to keep it simple, apply instance normalization followed by adjusting the mean to the corresponding historical average to keep it rational. We tune the initialization values of the linear weighting mechanism to prevent overfitting or reduce learning difficulty. To use Time-MMD dataset for TSF task, MM-TSFlib constrains the latest end date of the input text sequence to be earlier than the latest end date of the input sequence, in order to avoid information leakage. Overall, MM-TSFlib supports multimodal extensions of over 20 unimodal TSF algorithms via 7 open-source LLM models.

## J.1   List of Supported TSF

So far, our models supports following TSF models: TimeMixer (61), TSMixer (9), iTransformer (42), PatchTST (48), TimesNet (66), DLinear (74), LightTS (76), ETSformer (64), Non-stationary Transformer (44), FEDformer (82), Pyraformer (41), Autoformer (67), Informer (80), Reformer (32), Transformer (59), Mamba (23), SegRNN (36), Koopa (43), FreTS (72), TiDE (12), FiLM (81), MICN (60), Crossformer (78), TFT (35).

---

[28]https://github.com/thuml/Time-Series-Library
[29]https://huggingface.co/models

## J.2 List of Supported LLM

For LLM, MM-TSFlib supports BERT (14), GPT-2 (50) (Small, Medium, Large, Extra-Large), Llama-2-7B (58), and Llama-3-8B[30].

# K More Details of Experimental Setup

## K.1 Time-Series Forecasting Backbones

We deployed two sets of experiments upon TSF models: (1) Transformer-based, including Transformer (59), Reformer (32), Informer (80), Autoformer (67), Crossformer (78), Non-stationary Transformer (44),FEDformer (82), iTransformer (42). (2) MLP-based: DLinear (74). (3) Agnostic: FiLM (81). (4) LLM-based: Time-LLM (25).

- **Transformer**, a classic sequence-to-sequence model basing on multi-head attention mechanism.
- **Reformer**, an computational-efficient Transformer with advancements in attention hashing and reversible residual layers
- **Informer**, an advanced Transformer designed to tackle long-term forecasting problem with sparse attention layers and self-attention distilling.
- **Autoformer**, a Transformer-based model that keeps encoder-decoder structure but alters attention computations by auto-correlation mechanism in order to benefit long-term forecasting.
- **Crossformer**, a multi-variate Transformer-based model that explicitly explores and utilizes cross-dimension dependencies.
- **Non-stationary Transformer**, a Transformer that is designed to capture non-stationarity patterns instead of temporal correlation.
- **FEDformer**, a Transformer that explicitly use Fourier decomposition results to enhance long-term forecasting ability.
- **iTransformer**, a inverted Transformer that tokenizes multivariate time-series upon each timestamps/
- **DLinear**, a linear model that performs forecasting by a direct regression upon historical time series with a one-layer linear model.
- **FiLM**, a model-agnostic method that introduces Legendre and Fourier projections to denoise series and approximate historical information.
- **Time-LLM**, a framework that integrates LLM for time-series forecasting by reprogramming input series and then aligning with text prototypes.

## K.2 LLM Backbones

We use GPT-2 Small (50) for the majority of experiments, while other GPT-2 models and BERT, Llama-2, and Llama-3 are used in the ablation study.

- **BERT** a bidirectional transformer pre-trained on large text corpora with tasks like masked language modeling and next sentence prediction
- **GPT-2** an advanced language model that generates coherent and contextually relevant text by predicting subsequent words in a sentence, pre-trained on diverse internet text and capable of performing a variety of language tasks without task-specific fine-tuning
- **Llama-2** an accessible, open-source LLM designed to generate coherent and contextually relevant text by leveraging advanced transformer-based architecture
- **Llama-3** the latest iteration of the Llama model, offering enhanced text generation and comprehension abilities, further advancing the performance and versatility of its predecessors

---

[30]https://llama.meta.com/llama3

### K.3 Evaluation Metrics

We use mean squared error (MSE) as the evaluation metric for all experiments. The MSE is defined as the average of the squares of the errors. The error here is the difference between the forecasted value $y_i$ and the actual value $\hat{y}_i$. That is:

$$\text{MSE} = \frac{1}{n} \sum_{i=1}^{n} (y_i - \hat{y}_i)^2 \tag{1}$$

### K.4 Details of Implementation

We follow the commonly adopted setup for defining the forecasting horizon window length, as outlined in prior works (67; 66; 48). Specifically, for daily reported datasets, the forecasting horizon windows are chosen from the set [48, 96, 192, 336], with a fixed lookback window size of 96 and a consistent label window size of 48 for the decoder. Similarly, for the weekly reported dataset, we employ forecasting horizon windows from [12, 24, 36, 48], with a fixed lookback window size of 36 and a constant label window size of 18 for the decoder. Besides, for the monthly reported dataset, we employ forecasting horizon windows from [6,8,10,12], with a fixed lookback window size of 8 and a constant label window size of 4 for the decoder.

To avoid the time consumption of matching data one by one during training, we adopted a simple preprocessing method: tracing back the most recent k text samples based on the timestamp of the numerical sample to construct pre-stored pairs. We focused on the search text and retained the source, i.e., URL, to provide rich and formatted information. It is worth noting that this processing method is a simple preliminary exploration, and we look forward to more effective and efficient matching approaches tailored for time series forecasting tasks with our Time-MMD.

## L  Additional Experiments on Text Integration Approaches

The previously proposed multimodal integration approaches (25; 6) are designed specifically for prompt-based text rather than exogenous textual series. Specifically, the text used in these models serves as prompts, such as task descriptions, rather than our exogenous textual series. Prompts are static, fixed in length, highly templated, and do not provide additional information, which makes them significantly different from the textual series used in our Time-MMD. We choose Time-LLM (25), an advanced model in this category, as a representative. The experimental results are shown in Table 5.

| Model | Health | Energy | Traffic |
|---|---|---|---|
| TimeLLM (Unimodal) | 1.92 | 0.30 | 0.23 |
| TimeLLM (Multimodal_TimeLLM) | 2.51 | 0.39 | 0.27 |
| TimeLLM (Multimodal_Our) | 1.42 | 0.25 | 0.22 |

Table 5: Results of Additional Experiments on Multimodal Integration Approaches

The results show that the 'prompt as prefix' approach in TimeLLM, referred to as Multimodal_TimeLLM, is not suitable for handling textual series. Possible reasons include the longer length of the text series potentially overwhelming the numerical series information, among others. In contrast, our framework, although simple, proves to be effective."

We also empirically found that using only the embedding layer of the LLM to model text often achieves results close to or even better than the last layer. This is consistent with recent research findings (57), validating that there is still significant room for improvement when using LLMs for time series forecasting.

## M  Additional Experiments on Text Modeling Approaches

Results are provided in Table 6.

| Dataset | Health | Energy | Traffic |
|---|---|---|---|
| Reformer (Uni) | 1.85 | 0.43 | 0.31 |
| Reformer (Multi_Bert) | 1.27 | 0.40 | 0.17 |
| Reformer (Multi_Doc2Vec) | 1.35 | 0.43 | 0.22 |
| Informer (Uni) | 1.53 | 0.33 | 0.28 |
| Informer (Multi_Bert) | 1.22 | 0.29 | 0.17 |
| Informer (Multi_Doc2Vec) | 1.32 | 0.31 | 0.21 |

Table 6: Results of Additional Text Modeling Solutions

# N    Additional Experiments on Multimodal Modeling Approaches

## N.1    Using attention mechanisms

We use the output of the time series model to calculate attention scores for each token in the LLM output and perform weighted aggregation. For detailed implementation, please refer to our library. Results are provided in Table 7. This basic attention method has not yet demonstrated a clear advantage over the current simple combing approach. We look forward to future method designs.

| Dataset | Health | Energy | Traffic |
|---|---|---|---|
| Reformer (Multi_Our) | 1.27 | 0.41 | 0.17 |
| Reformer (Multi_Att) | 1.26 | 0.39 | 0.22 |
| Informer (Multi_Our) | 1.22 | 0.29 | 0.17 |
| Informer (Multi_Att) | 1.23 | 0.29 | 0.16 |

Table 7: Results of Using Attention Mechanisms for Multimodal TSF

## N.2    Using closed-source LLMs

We first used closed-source LLMs, such as GPT-3.5, to generate text-based predictions, and then encoded them using BERT. For detailed implementation, please check our library. Results are provided in Table 8. We initially observed that fully frozen closed-source LLMs are less effective than fine-tuned open-source LLM.

| Dataset | Health | Energy | Traffic |
|---|---|---|---|
| Reformer (Uni) | 1.86 | 0.43 | 0.31 |
| Reformer (Multi_Bert) | 1.27 | 0.40 | 0.17 |
| Reformer (Multi_GPT3.5) | 1.38 | 0.46 | 0.19 |
| Informer (Uni) | 1.53 | 0.33 | 0.28 |
| Informer (Multi_Bert) | 1.22 | 0.28 | 0.17 |
| Informer (Multi_GPT3.5) | 1.33 | 0.32 | 0.23 |

Table 8: Results of Using Closed-Source LLMs for Multimodal TSF

In addition, we empirically observe that using the encoder layers of LLMs usually yields better performance.

## O  Detailed Results

| Model | Modal | Horizon Window Length | | | |
|---|---|---|---|---|---|
| | | 6 | 8 | 10 | 12 |
| FiLM | Uni | 0.07 | 0.09 | 0.12 | 0.15 |
| | Multi | 0.06 | 0.09 | 0.10 | 0.14 |
| DLinear | Uni | 0.11 | 0.24 | 0.18 | 0.23 |
| | Multi | 0.09 | 0.17 | 0.13 | 0.16 |
| Transformer | Uni | 0.24 | 0.35 | 0.40 | 0.50 |
| | Multi | 0.14 | 0.16 | 0.24 | 0.24 |
| Reformer | Uni | 0.38 | 0.26 | 0.51 | 0.70 |
| | Multi | 0.19 | 0.18 | 0.27 | 0.32 |
| Informer | Uni | 0.51 | 0.61 | 0.66 | 0.80 |
| | Multi | 0.16 | 0.21 | 0.34 | 0.34 |
| Autoformer | Uni | 0.08 | 0.09 | 0.11 | 0.14 |
| | Multi | 0.08 | 0.09 | 0.10 | 0.13 |
| FEDformer | Uni | 0.06 | 0.08 | 0.10 | 0.13 |
| | Multi | 0.06 | 0.07 | 0.10 | 0.13 |
| Nonstationary Transformer | Uni | 0.06 | 0.07 | 0.10 | 0.12 |
| | Multi | 0.05 | 0.07 | 0.09 | 0.12 |
| Crossformer | Uni | 0.31 | 0.35 | 0.38 | 0.46 |
| | Multi | 0.11 | 0.16 | 0.20 | 0.26 |
| PatchTST | Uni | 0.06 | 0.08 | 0.10 | 0.13 |
| | Multi | 0.06 | 0.08 | 0.10 | 0.13 |
| iTransformer | Uni | 0.06 | 0.08 | 0.10 | 0.13 |
| | Multi | 0.06 | 0.08 | 0.10 | 0.13 |
| Time-LLM | Uni | 0.08 | 0.09 | 0.10 | 0.14 |
| | Multi | 0.06 | 0.08 | 0.10 | 0.13 |

Table 9: Agriculture

| Model | Modal | Horizon Window Length 6 | 8 | 10 | 12 |
|---|---|---|---|---|---|
| FiLM | Uni | 1.42 | 1.39 | 1.40 | 1.40 |
| | Multi | 1.15 | 1.15 | 1.14 | 1.17 |
| DLinear | Uni | 1.35 | 1.41 | 1.36 | 1.36 |
| | Multi | 1.06 | 1.05 | 1.07 | 1.08 |
| Transformer | Uni | 1.04 | 1.14 | 1.12 | 1.11 |
| | Multi | 0.97 | 1.01 | 1.00 | 1.00 |
| Reformer | Uni | 1.24 | 1.06 | 1.13 | 1.16 |
| | Multi | 0.97 | 0.95 | 0.94 | 0.98 |
| Informer | Uni | 1.08 | 1.11 | 1.08 | 1.07 |
| | Multi | 1.04 | 1.03 | 1.04 | 1.02 |
| Autoformer | Uni | 1.30 | 1.24 | 1.28 | 1.25 |
| | Multi | 1.08 | 1.02 | 1.05 | 1.05 |
| FEDformer | Uni | 1.32 | 1.36 | 1.28 | 1.27 |
| | Multi | 0.98 | 1.00 | 1.03 | 1.02 |
| Nonstationary Transformer | Uni | 1.30 | 1.32 | 1.36 | 1.32 |
| | Multi | 1.00 | 1.02 | 1.02 | 1.01 |
| Crossformer | Uni | 1.12 | 1.10 | 1.12 | 1.10 |
| | Multi | 1.00 | 0.99 | 1.00 | 1.01 |
| PatchTST | Uni | 1.36 | 1.33 | 1.27 | 1.28 |
| | Multi | 0.99 | 1.01 | 1.04 | 1.06 |
| iTransformer | Uni | 1.16 | 1.23 | 1.24 | 1.22 |
| | Multi | 0.99 | 1.01 | 1.04 | 1.06 |
| Time-LLM | Uni | 1.36 | 1.26 | 1.27 | 1.27 |
| | Multi | 0.99 | 1.01 | 1.04 | 1.07 |

Table 10: Climate

| | Horizon Window Length | 6 | 8 | 10 | 12 |
|---|---|---|---|---|---|
| Model | Modal | | | | |
| FiLM | Uni | 0.04 | 0.04 | 0.03 | 0.04 |
| | Multi | 0.03 | 0.03 | 0.03 | 0.03 |
| DLinear | Uni | 0.08 | 0.18 | 0.10 | 0.11 |
| | Multi | 0.04 | 0.10 | 0.03 | 0.03 |
| Transformer | Uni | 0.60 | 0.69 | 1.19 | 1.12 |
| | Multi | 0.14 | 0.31 | 0.23 | 0.36 |
| Reformer | Uni | 0.65 | 0.76 | 1.02 | 0.55 |
| | Multi | 0.22 | 0.21 | 0.32 | 0.28 |
| Informer | Uni | 1.59 | 1.82 | 1.74 | 2.08 |
| | Multi | 0.28 | 0.36 | 0.49 | 0.50 |
| Autoformer | Uni | 0.08 | 0.10 | 0.07 | 0.06 |
| | Multi | 0.07 | 0.08 | 0.07 | 0.07 |
| FEDformer | Uni | 0.05 | 0.06 | 0.05 | 0.06 |
| | Multi | 0.04 | 0.03 | 0.04 | 0.03 |
| Nonstationary Transformer | Uni | 0.02 | 0.02 | 0.02 | 0.03 |
| | Multi | 0.02 | 0.02 | 0.02 | 0.02 |
| Crossformer | Uni | 0.95 | 1.06 | 1.22 | 1.24 |
| | Multi | 0.17 | 0.16 | 0.27 | 0.29 |
| PatchTST | Uni | 0.02 | 0.02 | 0.02 | 0.02 |
| | Multi | 0.02 | 0.02 | 0.02 | 0.02 |
| iTransformer | Uni | 0.02 | 0.02 | 0.02 | 0.02 |
| | Multi | 0.01 | 0.02 | 0.02 | 0.02 |
| Time-LLM | Uni | 0.06 | 0.08 | 0.02 | 0.05 |
| | Multi | 0.02 | 0.02 | 0.02 | 0.02 |

Table 11: Economy

| | Horizon Window Length | 12 | 24 | 36 | 48 |
|---|---|---|---|---|---|
| Model | Modal | | | | |
| FiLM | Uni | 0.21 | 0.30 | 0.37 | 0.49 |
| | Multi | 0.17 | 0.28 | 0.36 | 0.48 |
| DLinear | Uni | 0.26 | 0.32 | 0.39 | 0.50 |
| | Multi | 0.22 | 0.29 | 0.36 | 0.47 |
| Transformer | Uni | 0.18 | 0.26 | 0.36 | 0.44 |
| | Multi | 0.13 | 0.22 | 0.32 | 0.42 |
| Reformer | Uni | 0.28 | 0.38 | 0.49 | 0.57 |
| | Multi | 0.25 | 0.38 | 0.43 | 0.54 |
| Informer | Uni | 0.18 | 0.29 | 0.35 | 0.48 |
| | Multi | 0.15 | 0.24 | 0.32 | 0.44 |
| Autoformer | Uni | 0.18 | 0.31 | 0.34 | 0.47 |
| | Multi | 0.16 | 0.27 | 0.32 | 0.45 |
| FEDformer | Uni | 0.11 | 0.24 | 0.34 | 0.45 |
| | Multi | 0.09 | 0.21 | 0.32 | 0.44 |
| Nonstationary Transformer | Uni | 0.11 | 0.21 | 0.34 | 0.48 |
| | Multi | 0.10 | 0.20 | 0.28 | 0.46 |
| Crossformer | Uni | 0.14 | 0.29 | 0.36 | 0.41 |
| | Multi | 0.13 | 0.26 | 0.36 | 0.41 |
| PatchTST | Uni | 0.10 | 0.21 | 0.30 | 0.42 |
| | Multi | 0.10 | 0.21 | 0.29 | 0.41 |
| iTransformer | Uni | 0.10 | 0.21 | 0.30 | 0.42 |
| | Multi | 0.09 | 0.19 | 0.29 | 0.41 |
| Time-LLM | Uni | 0.16 | 0.27 | 0.31 | 0.45 |
| | Multi | 0.10 | 0.20 | 0.29 | 0.41 |

Table 12: Energy

| | Horizon Window Length | 48 | 96 | 192 | 336 |
|---|---|---|---|---|---|
| Model | Modal | | | | |
| FiLM | Uni | 0.32 | 0.35 | 0.35 | 0.32 |
| | Multi | 0.30 | 0.32 | 0.32 | 0.30 |
| DLinear | Uni | 0.41 | 0.57 | 0.73 | 0.59 |
| | Multi | 0.32 | 0.40 | 0.46 | 0.42 |
| Transformer | Uni | 0.32 | 0.32 | 0.48 | 0.44 |
| | Multi | 0.59 | 0.61 | 0.70 | 0.32 |
| Reformer | Uni | 0.39 | 0.45 | 0.51 | 0.48 |
| | Multi | 0.29 | 0.35 | 0.36 | 0.32 |
| Informer | Uni | 0.39 | 0.42 | 0.46 | 0.48 |
| | Multi | 0.31 | 0.33 | 0.39 | 0.34 |
| Autoformer | Uni | 0.43 | 0.36 | 0.52 | 0.37 |
| | Multi | 0.35 | 0.35 | 0.35 | 0.34 |
| FEDformer | Uni | 0.36 | 0.43 | 0.42 | 0.35 |
| | Multi | 0.30 | 0.34 | 0.34 | 0.33 |
| Nonstationary Transformer | Uni | 0.31 | 0.39 | 0.43 | 0.38 |
| | Multi | 0.29 | 0.31 | 0.32 | 0.30 |
| Crossformer | Uni | 0.34 | 0.33 | 0.73 | 0.53 |
| | Multi | 0.29 | 0.30 | 0.36 | 0.36 |
| PatchTST | Uni | 0.35 | 0.38 | 0.36 | 0.32 |
| | Multi | 0.31 | 0.32 | 0.32 | 0.30 |
| iTransformer | Uni | 0.28 | 0.29 | 0.30 | 0.28 |
| | Multi | 0.28 | 0.29 | 0.29 | 0.27 |
| Time-LLM | Uni | 0.38 | 0.37 | 0.45 | 0.33 |
| | Multi | 0.29 | 0.30 | 0.31 | 0.28 |

Table 13: Environment

| Model | Horizon Window Length Modal | 12 | 24 | 36 | 48 |
|---|---|---|---|---|---|
| FiLM | Uni | 2.53 | 2.59 | 2.46 | 2.38 |
| | Multi | 1.67 | 1.83 | 1.80 | 1.81 |
| DLinear | Uni | 2.37 | 2.61 | 2.50 | 2.48 |
| | Multi | 1.62 | 1.67 | 1.68 | 1.78 |
| Transformer | Uni | 1.22 | 1.56 | 1.43 | 1.55 |
| | Multi | 0.93 | 1.34 | 1.26 | 1.29 |
| Reformer | Uni | 1.63 | 1.99 | 1.91 | 1.90 |
| | Multi | 1.06 | 1.30 | 1.33 | 1.39 |
| Informer | Uni | 1.24 | 1.61 | 1.61 | 1.67 |
| | Multi | 0.98 | 1.23 | 1.28 | 1.40 |
| Autoformer | Uni | 1.99 | 2.25 | 2.26 | 2.39 |
| | Multi | 1.43 | 1.74 | 1.76 | 1.69 |
| FEDformer | Uni | 1.08 | 1.58 | 1.69 | 1.76 |
| | Multi | 0.92 | 1.25 | 1.36 | 1.42 |
| Nonstationary Transformer | Uni | 1.19 | 1.68 | 1.91 | 2.02 |
| | Multi | 0.94 | 1.14 | 1.17 | 1.30 |
| Crossformer | Uni | 1.45 | 1.57 | 1.62 | 1.65 |
| | Multi | 1.01 | 1.29 | 1.28 | 1.37 |
| PatchTST | Uni | 1.23 | 1.63 | 1.78 | 1.86 |
| | Multi | 0.98 | 1.27 | 1.49 | 1.60 |
| iTransformer | Uni | 1.14 | 1.62 | 1.84 | 1.89 |
| | Multi | 0.97 | 1.38 | 1.71 | 1.72 |
| Time-LLM | Uni | 1.60 | 1.94 | 1.95 | 2.17 |
| | Multi | 0.98 | 1.36 | 1.65 | 1.69 |

Table 14: Health(US)

| | Horizon Window Length | 6 | 8 | 10 | 12 |
|---|---|---|---|---|---|
| Model | Modal | | | | |
| FiLM | Uni | 127.92 | 122.53 | 125.10 | 127.06 |
| | Multi | 105.92 | 107.25 | 109.80 | 110.94 |
| DLinear | Uni | 107.12 | 111.72 | 114.42 | 116.65 |
| | Multi | 104.28 | 106.76 | 108.53 | 109.90 |
| Transformer | Uni | 128.42 | 130.78 | 131.65 | 133.04 |
| | Multi | 122.54 | 126.66 | 125.95 | 127.46 |
| Reformer | Uni | 124.48 | 125.68 | 126.51 | 127.26 |
| | Multi | 122.11 | 123.85 | 124.54 | 125.01 |
| Informer | Uni | 129.43 | 131.43 | 132.84 | 133.54 |
| | Multi | 121.81 | 124.24 | 126.98 | 126.73 |
| Autoformer | Uni | 125.67 | 119.82 | 119.72 | 126.67 |
| | Multi | 107.95 | 114.98 | 113.34 | 116.33 |
| FEDformer | Uni | 114.98 | 112.14 | 117.92 | 121.90 |
| | Multi | 109.28 | 109.55 | 115.99 | 117.22 |
| Nonstationary Transformer | Uni | 105.77 | 106.90 | 109.72 | 111.16 |
| | Multi | 104.15 | 105.99 | 110.49 | 110.69 |
| Crossformer | Uni | 124.70 | 124.97 | 127.49 | 128.47 |
| | Multi | 120.99 | 122.40 | 124.34 | 125.50 |
| PatchTST | Uni | 107.31 | 111.42 | 114.90 | 113.11 |
| | Multi | 108.75 | 111.26 | 114.61 | 114.67 |
| iTransformer | Uni | 113.20 | 115.26 | 116.32 | 117.59 |
| | Multi | 114.58 | 116.49 | 116.25 | 116.31 |
| Time-LLM | Uni | 112.96 | 116.53 | 116.46 | 119.60 |
| | Multi | 110.86 | 114.54 | 115.56 | 115.63 |

Table 15: Security

| Model | Horizon Window Length / Modal | 6 | 8 | 10 | 12 |
|---|---|---|---|---|---|
| FiLM | Uni | 1.03 | 1.07 | 1.16 | 1.24 |
| | Multi | 0.87 | 0.97 | 1.05 | 1.06 |
| DLinear | Uni | 1.01 | 1.13 | 1.20 | 1.24 |
| | Multi | 0.91 | 1.02 | 1.14 | 1.15 |
| Transformer | Uni | 0.78 | 0.93 | 0.98 | 1.02 |
| | Multi | 0.73 | 0.84 | 0.91 | 0.90 |
| Reformer | Uni | 0.81 | 0.90 | 0.96 | 1.05 |
| | Multi | 0.80 | 0.85 | 0.95 | 1.03 |
| Informer | Uni | 0.77 | 0.86 | 0.93 | 0.94 |
| | Multi | 0.74 | 0.79 | 0.86 | 0.85 |
| Autoformer | Uni | 0.89 | 1.05 | 1.05 | 1.17 |
| | Multi | 0.88 | 0.96 | 1.04 | 1.12 |
| FEDformer | Uni | 0.81 | 0.92 | 1.02 | 1.11 |
| | Multi | 0.80 | 0.91 | 0.98 | 1.04 |
| Nonstationary Transformer | Uni | 0.85 | 1.17 | 1.29 | 1.31 |
| | Multi | 0.78 | 1.08 | 1.16 | 1.23 |
| Crossformer | Uni | 0.79 | 0.83 | 0.97 | 0.94 |
| | Multi | 0.75 | 0.83 | 0.89 | 0.93 |
| PatchTST | Uni | 0.88 | 0.99 | 1.07 | 1.18 |
| | Multi | 0.82 | 0.92 | 1.02 | 1.10 |
| iTransformer | Uni | 1.03 | 1.11 | 1.21 | 1.22 |
| | Multi | 0.90 | 1.08 | 1.12 | 1.15 |
| Time-LLM | Uni | 0.90 | 1.00 | 1.07 | 1.19 |
| | Multi | 0.90 | 0.99 | 1.08 | 1.10 |

Table 16: SocialGood

| | Horizon Window Length | 6 | 8 | 10 | 12 |
|---|---|---|---|---|---|
| Model | Modal | | | | |
| FiLM | Uni | 0.28 | 0.28 | 0.27 | 0.31 |
| | Multi | 0.25 | 0.24 | 0.24 | 0.29 |
| DLinear | Uni | 0.35 | 0.42 | 0.34 | 0.38 |
| | Multi | 0.28 | 0.30 | 0.25 | 0.31 |
| Transformer | Uni | 0.29 | 0.29 | 0.29 | 0.29 |
| | Multi | 0.15 | 0.15 | 0.16 | 0.18 |
| Reformer | Uni | 0.31 | 0.28 | 0.31 | 0.33 |
| | Multi | 0.16 | 0.16 | 0.17 | 0.19 |
| Informer | Uni | 0.26 | 0.27 | 0.27 | 0.30 |
| | Multi | 0.15 | 0.16 | 0.17 | 0.18 |
| Autoformer | Uni | 0.20 | 0.20 | 0.20 | 0.29 |
| | Multi | 0.19 | 0.19 | 0.20 | 0.27 |
| FEDformer | Uni | 0.21 | 0.21 | 0.20 | 0.27 |
| | Multi | 0.16 | 0.16 | 0.17 | 0.23 |
| Nonstationary Transformer | Uni | 0.19 | 0.21 | 0.20 | 0.27 |
| | Multi | 0.19 | 0.20 | 0.19 | 0.25 |
| Crossformer | Uni | 0.25 | 0.24 | 0.25 | 0.26 |
| | Multi | 0.17 | 0.16 | 0.17 | 0.19 |
| PatchTST | Uni | 0.21 | 0.21 | 0.22 | 0.28 |
| | Multi | 0.18 | 0.19 | 0.20 | 0.27 |
| iTransformer | Uni | 0.20 | 0.21 | 0.22 | 0.28 |
| | Multi | 0.19 | 0.20 | 0.21 | 0.27 |
| Time-LLM | Uni | 0.21 | 0.21 | 0.21 | 0.27 |
| | Multi | 0.19 | 0.20 | 0.20 | 0.27 |

Table 17: Traffic

## P  Dataset Documentations

The dataset is provided in `csv` format.

Each numerical data point contains the following attribute:

- start time
- end time
- values of target variables
- values of other variables

Each textual data point contains the following attribute:

- start time
- end time
- extracted fact text (content & data source)
- extracted prediction text (content & data source)

## Q  Intended Uses

Time-MMD is intended for researchers in machine learning and related fields to develop novel methods for multimodal time-series analysis. We also aim to help developers to train or fine-tune multi-modal foundation time-series models using our dataset.

## R  Accessing

Time-MMD is publicly available in `https://github.com/AdityaLab/Time-MMD`. A detailed demo on how the dataset can be read and used can be found at `https://github.com/AdityaLab/MM-TSFlib`. A Croissant card of Time-MMD can be found in Google Drive.

## S  Statement

The authors will bear all responsibility in case of violation of rights.

## T  Hosting and Maintenance Plan

Time-MMD is hosted and version-tracked via GitHub. It will be permanently available under the link `https://github.com/AdityaLab/Time-MMD`. The download link of all the datasets can be found in the GitHub repository. We plan to regularly update the cutoff time of the data in Time-MMD at a frequency of every 6 months.

## U  Time-MMD Community

Time-MMD is a community-driven and open-source initiative. We are committed and have resources to maintain and actively develop Time-MMD in the future. We plan to grow Time-MMD to include more languages, domains, and multimodal time series analysis tasks. We welcome external contributors.

## V  Licensing

We license our dataset by ODC-By v1.0. While parsing the data, we follow the terms of both Google Search API[31] and WebScraper[32]. We further make sure that all data sources obtained in Time-MMD

---

[31]`https://developers.google.com/terms`
[32]`https://webscraper.io/extension-privacy-policy`

are in public domain and authorized for non-commercial distribution. The authors will bear all responsibility in case of violation of rights.

