# OpenReview forum: "Time-MMD: Multi-Domain Multimodal Dataset for Time Series Analysis"
_NeurIPS.cc/2024/Datasets_and_Benchmarks_Track — NeurIPS 2024 Track Datasets and Benchmarks Poster_

### Official Review · Reviewer_otUR · 2024-06-28
**Good and Useful Benchmark Dataset**

**Rating:** 6
**Confidence:** 4
**Clarity:** Overall it is clear, but some parts n…

**Review:**

This paper is overall well-written and easy to follow, and the proposed dataset and the library are necessary for the research community. they will support the development of the new research domain "LLM for time series".

However, this paper still needs improvement in terms of some misleading statements, unclear information in some parts, and poor documentation, which will mentioned below.

**Strengths:**

1. This paper is overall clearly written and easy to follow.
2. The datasets and the library are of great significance to the research community.

**Additional Feedback:**

Please refer to the above weaknesses and questions.

**Correctness:**

Some claims seem misleading. For example, this paper focuses on time series forecasting, but the title mentions time series analysis, which includes time series anomaly detection, classification, and missing value imputation, which are missing in the paper. Similar statements are In the boarder impact section, see points 5 and 6 in "Opportunities For Improvement".

**Documentation:**

Unfortunately, the doucmentation needs improvement because it lacks example usages and explanations of the datasets and the library.

**Ethics:**

No ethical concern.

**Limitations:**

Unfortunately, this paper does not mention the limitations of the paper, which should be addressed.

**Opportunities For Improvement:**

This paper suffers from the following limitations:
1. Some claims seem exaggerated. This paper focuses on the time series forecasting task, but the title is "time series analysis", which may be misleading.
2. In Table 1, some statistics of datasets should be mentioned, such as the number of samples and features if only.
3. It is unclear how to align time series data and text data, and it is unclear whether there is any frequency difference between the two modalities.
4. It is unclear how to ensure that the collected text data is related to time series data. For example, is there any way to explain the change of time series data in terms of text data?
5. In the boarder impacts section, it is unclear how to adapt the proposed time-MMD dataset for the anomaly detection task. Specifically, it is unclear how to determine the anomalies in the time-MMD dataset.
6. Also in the boarder impacts section, the authors mentioned the proposed time-MMD dataset supports time series foundation models. However such models require a larger training corpus. It is unclear whether the proposed time-MMD dataset is large enough to support this.
7. The documentations need improvement. It is difficult to use the library and datasets due to a lack of explanations on the example usages and dataset introduction.

**Relation To Prior Work:**

To my understanding, this paper is the first one for multi-modal time series forecasting.

**Summary And Contributions:**

This paper proposes Time-MMD, the first benchmark dataset for multi-modal multi-domain time series dataset covering 9 primary data domains. This paper also proposes MM_TSFib, a new library for multi-modal time series forecasting. The experimental results demonstrate the improvement of multi-modal time series forecasting over unimodal scenarios.

---

> ### Author Rebuttal · Authors · 2024-08-17
>
> We sincerely appreciate the reviewer's insightful comments and detailed suggestions.
>
> **Response to ”1. Some claims seem exaggerated”**
>
> We appreciate the reviewer's constructive suggestions. Time-MMD now directly supports **the most common short-term forecasting, long-term forecasting and additional imputation and anomaly detection tasks**. For imputation, we generate missing values following mainstream methods, released at our repo https://github.com/AdityaLab/Time-MMD. For anomaly detection, we have provided preliminary anomaly labels, available at https://anonymous.4open.science/r/Time-MMD-Anomaly-9424/
> We plan to release it after manual verification. We are dedicated to making Time-MMD support multiple tasks. We are willing to consider modifying the title to avoid potential misleading in the final version.
>
> **Response to “2. Table 1 is not clear"**
>
> We appreciate the reviewer's constructive suggestions. We would like to provide the following clarifications:
>
> 1. Regarding the number of features: This work focuses on univariate forecasting, so the number of target variables is 1 in each dataset. However, we do provide covariates in some datasets. We will include this in the final version.
> 2. Regarding the number of samples: In Table 1, "Timestamps" refers to the number of samples. We will replace this with "Number of Samples" in the final version.
>
> **Response to “3.1 unclear about temporal alignment"**
>
> We labeled the start time and end time for both text and numerical data, enabling different alignment requirements, as detailed in Section 2.3. This meets the needs of different tasks, such as the most precise temporal matching or simply a temporal overlap, and also prevents information leakage. The effectiveness of temporal alignment is verified by “Coverage” in Table 2.
>
> **Response to “3.2 unclear about frequency difference"**
>
> We appreciate the reviewer's constructive suggestions.  We provide **multiple frequencies** of text sequences, as detailed in Section 2.2 and Appendix J. Specifically, the frequency of keyword search text is daily, which **ensures the most precise frequency alignment**. Additionally, the frequency of the selected reports is diverse. For example, in the Health domain, numerical series are weekly, while text sequences cover daily search texts, weekly reports, quarterly reports, and annual reports. We combined these two types of text sources to ensure coverage for the numerical data, as detailed in Table 2. We will include this discussion in the final version.

---

> > ### Author Rebuttal · Authors · 2024-08-17
> >
> > **Response to “4. unclear how to ensure semantic alignment”**
> >
> > We appreciate the reviewer's constructive suggestions. We want to first point out that ensuring semantic alignment is a longstanding challenge in constructing multimodal time-series datasets. Previous works [1-2], focused  on the finance domain, simply used general financial news to match a specific stock price.
> >
> > In Time-MMD, we explore a more reasonable construction procedure, trying our best to ensure semantic alignment. Specifically, we **ensured relevance through careful selection of data sources, including customized keyword web searches and report series selections, followed by filtering via LLMs**, as detailed in Section 2.2.
> >
> > The effectiveness of LLM filtering is detailed in Table 2 and Appendix B. We further provide **a small-scale manual data check** in Appendix G, which verify the consistency between human and LLM judgments on relevance. Following the reviewer's suggestion, we have also provided a case **example** of text explaining data changes in the **attached PDF**.
> >
> > [1] Dong, Zihan, Xinyu Fan, and Zhiyuan Peng. "FNSPID: A Comprehensive Financial News Dataset in Time Series." arXiv preprint arXiv:2402.06698 (2024).
> >
> > [2] Saeede Anbaee Farimani, M. V. Jahan, A. M. Fard, and Gholamreza Haffari. 2021. Leveraging Latent EcoNomic Concepts and Sentiments in the News for Market Prediction
> >
> > **Response to "5. how to support anomaly detection”**
> >
> > We appreciate the reviewer's insightful question.
> >
> > Firstly, we would like to point out that annotating anomalies is a core challenge in anomaly detection area [1]. The construction of Time-MMD provides a helpful initial dataset for the anomaly detection field.
> >
> > Specifically, Time-MMD provides a reasonable approach to **utilize text data for anomaly labeling**. For example, in influenza (health) dataset, if "flu outbreak" is mentioned in the text data, we can initially label the aligned numerical data as anomalous. We have provided an **initial labeled dataset** at https://anonymous.4open.science/r/Time-MMD-Anomaly-9424/, and we plan to release it after manual verification. We will provide specific implementation details in the final version of our paper.
> >
> > [1] Wu, Renjie, and Eamonn J. Keogh. "Current time series anomaly detection benchmarks are flawed and are creating the illusion of progress." IEEE transactions on knowledge and data engineering 35.3 (2021): 2421-2429.
> >
> > **Response to "6. unclear whether large enough to support training foundation model”**
> >
> > We appreciate the reviewer's insightful question. We would like to first clarify that our plan is to support multimodal foundation models based on foundational language models and time-series models [1-3], which is a verified approach in CV+NLP [4].
> >
> > Secondly, the question "whether large is enough" is almost unable to answer at this stage. Given that the size of foundational time-series models is typically small, such as 8M [1], we would like to draw on the experience of fine-tuning LLMs as a rough reference. A recent work [5] shows that **6.5K samples is the saturation point** for fine-tuning LLMs. In comparison, **Time-MMD contains over 17K samples**.
> >
> > To the best of our knowledge, Time-MMD **is the largest** multimodal time-series dataset across multiple domains. We commit to continuously updating the dataset to further expand its scale. We agree with the reviewer’s perspective that it is unclear at this time. Therefore, we are willing to change Section 5 from "Broader Impacts" to "Potential Future Work" and include more discussion about this in the final version.
> >
> > [1] Woo, Gerald, et al. "Unified Training of Universal Time Series Forecasting Transformers." Forty-first International Conference on Machine Learning.
> >
> > [2] Ansari, Abdul Fatir, et al. "Chronos: Learning the language of time series." arXiv preprint arXiv:2403.07815 (2024).
> >
> > [3] Das, Abhimanyu, et al. "A decoder-only foundation model for time-series forecasting." Forty-first International Conference on Machine Learning.
> >
> > [4] Li, Junnan, et al. "Blip-2: Bootstrapping language-image pre-training with frozen image encoders and large language models." International conference on machine learning. PMLR, 2023.
> >
> > [5] Oliver, Michael, and Guan Wang. "Crafting Efficient Fine-Tuning Strategies for Large Language Models." arXiv preprint arXiv:2407.13906 (2024).

---

> > > ### Author Rebuttal · Authors · 2024-08-17
> > >
> > > **Response to "7. documentations need improvement"**
> > >
> > >
> > > We appreciate the reviewer's constructive suggestions. We have **significantly improved the documentations** including detailed description, usage guidelines, and news list. We have also expanded the functionality of the dataset and library. Please check the **updated** dataset at https://github.com/AdityaLab/Time-MMD   and library at https://github.com/AdityaLab/MM-TSFlib

---

> > > > ### Comment · Reviewer_otUR · 2024-08-21
> > > > **Thank you for the rebuttal**
> > > >
> > > > I appreciate the efforts authors spare on the rebuttal. Time-MMD is a good contribution for the multi-modal time series analysis. Even though some extra work need to be done, I raise the score for encouragement.

---

> > > > > ### Author Rebuttal · Authors · 2024-08-24
> > > > >
> > > > > We thank the reviewer for recognizing our work, "Time-MMD is a good contribution to the multi-modal time series analysis." We also greatly appreciate the reviewer's suggestions for improvement in the final version.

---

### Official Review · Reviewer_mAU8 · 2024-07-24
**They propose a new multimodal and multi-domain time serious datasets and Library.**

**Rating:** 7
**Confidence:** 3
**Correctness:** The submission is correct and the dat…
**Clarity:** The paper is well-written and easy to…

**Review:**

1. **Coarse-grained modality alignment**. What is the task-aware alignment in Figure 1. If your just align start time and end time of numerical and textual information, I argue that these may not be a kind of Coarse-grained modality alignment.

2. **Concern about the Effectiveness of  Multimodal Dataset**. I am confused about the effectiveness of multimodal datasets for time serious forecasting. In Figure 6, you show the average performance of MSE. If the model have improved performance on all these datasets separately or decreased on some specific dataset? And if they decrease performance on some specific dataset, I would like to know why.

3. **Concern about the Differences of Multi-domain**. In Figure 2 and  Figure7 (a), we could see almost the same performance and similar patterns and trends of some domains. I am still not sure why authors select these domains for constructing their datasets.

4. **Need More Discussion for Experiments**. It is kind of strange that BERT can have better performance than llama3 in Figure7(b). I am also confused about why horizon window size is stable for baselines. In my understanding, a longer window size could bring more information to improve the performance.

**Strengths:**

1.  Pioneering Multi-Domain Multimodal Time-Series Dataset.

2.  Pilot Multimodal Time-Series Forecasting Library: MM-TSFlib.

3.  Extensive Evaluations with Significant Improvement.

**Additional Feedback:**

Please see Review above.

**Documentation:**

They are sufficient detail on data collection and organization. The datasets are ethical and responsible for using.

**Ethics:**

They discussed ethics in their study.

**Limitations:**

Authors discuss the limitations in their study.

**Opportunities For Improvement:**

1. Only numerical and textual modality used for dataset construction. Consider image and audio information maybe better.

2. Consider single variable and multiple variables would be better.

**Relation To Prior Work:**

They lack the discussion about the related work.

**Summary And Contributions:**

They propose a new multimodal and multi-domain time serious datasets and Library. This study introduces a new multi-domain and multimodal time series dataset, called Time-MMD. And they also develop the first multimodal time-series forecasting library for in-depth analyses. Extensive evaluations show the high quality of their datasets.

---

> ### Author Rebuttal · Authors · 2024-08-17
>
> We thank the reviewer for their insightful comments and recognition of our dataset, library and experiments.
>
> **Response to “Coarse-grained modality alignment.”**
>
> Our clarification is as follows: The task-aware alignment in Figure 1 is not coarse-grained modality alignment.
>
> Specifically, the alignment between text and numerical series involves two aspects: a) semantic alignment b) temporal alignment. The task-aware alignment refers to temporal alignment after the semantic alignment step.
>
> For the semantic alignment, we ensured relevance through careful selection of data sources, including customized keyword web searches and selected report series, followed by filtering via LLMs, as detailed in Section 2.2 and Appendix C. These designs ensure fine-grained semantic alignment, verified by “Relevance” in Table 2.
>
> For the temporal alignment, we labeled the start time and end time for both text and numerical data, enabling different task requirements, such as the most precise temporal matching or simply a temporal overlap. It is worth noting that the search text series are set at a daily frequency, ensuring the finest-grained temporal alignment with numerical series. The effectiveness of temporal alignment is verified by “Coverage” in Table 2.
>
> We would greatly appreciate it if the reviewer would like to further clarify whether "coarse-grained alignment" refers to semantic or temporal alignment.
>
> **Response to  “Concern about the Effectiveness of Multimodal Dataset”**
>
> Our clarification is as follows: we choose to present the averaged results of each model to save spaces in Figure 6 but also present the **average MSE reduction on each dataset** in Figure 7(a) and **full experiment results** in Appendix J. Discussions can be found in Section 4.2.
>
> According to our experimental results, **improvements were achieved across all datasets**, reducing the MSE by an average of over 20% and up to 40% in some domains with rich textual data. This widespread and significant improvement has demonstrated the effectiveness of our multimodal dataset.
>
> More specifically, in more than 1k experiments, the multimodal versions outperformed the corresponding unimodal versions in more than 95% of cases. The remaining 5% of cases may be due to the fact that our simple multimodal framework is not powerful enough.
>
> **Response to “Concern about the Differences of Multi-domain.”**
>
> We respectfully disagree with the reviewer's viewpoint. Our domains exhibit significant differences.
>
> 1. Performance shows differences. We would like to respectfully correct the reviewer: the y-axis in Figure 7(a) represents the percentage reduction in MSE, not the MSE itself. Thus, the performance in different domains is not similar, also verified by detailed MSE results in Appendix J. For example, although agriculture and health domains show similar percentage reductions, their MSE differ significantly: below 0.1 vs above 1.0.
>
> 2. Numerical data shows differences. We respectfully disagree with the reviewer's viewpoint regarding similar patterns and trends in Figure 2. Generally, 3 of 10 datasets show an upward trend, 1 shows a downward trend, and 6 have no clear trend. There is also no consistent periodicity. Additionally, Table 1 shows the differences in the statistical properties, including number of samples and frequency.
>
> 3. Textual data shows differences. Appendix D reveals clear differences in the statistics of text data, including richness (tokens) and availability (coverage), among other factors.
>
> We will include a discussion on domain differences in the final version. We appreciate the reviewer's insightful comments.

---

> > ### Comment · Reviewer_mAU8 · 2024-08-31
> >
> > Thanks for your response. I am happy to see this good work for our community.

---

> > > ### Author Rebuttal · Authors · 2024-09-01
> > >
> > > We thank the reviewer for the recognition of our work.

---

> ### Author Rebuttal · Authors · 2024-08-17
>
> **Response to “Not Sure about Domain Selection”**
>
> We thank the reviewer for their insightful comments. Our domain selection is based on the following three considerations:
>
> 1. Most of our selected domains are also widely used in existing time-series works[1-4] but lack textual and numerical alignment. We chose these domains to facilitate comparison by researchers, and all these domains have significant real-world importance.
>
> 2. The remaining three domains, including social good, agriculture, and security, are often overlooked by researchers. We included these domains to contribute to a more comprehensive dataset.
>
> 3. Additionally, we also considered underrepresented groups, as reflected in the Health (Africa) dataset, compared with the Health (U.S.).
>
> To the best of our knowledge, our Time-MMD is the most domain-inclusive multimodal time-series dataset. We are committed to continuously updating it to incorporate more domains. We will include a discussion on domain selection in the final version.
>
> [1] Liu, Yong, et al. "Timer: Generative Pre-trained Transformers Are Large Time Series Models." Forty-first International Conference on Machine Learning.
>
> [2] Ansari, Abdul Fatir, et al. "Chronos: Learning the language of time series." arXiv preprint arXiv:2403.07815 (2024).
>
> [3] Das, Abhimanyu, et al. "A decoder-only foundation model for time-series forecasting." Forty-first International Conference on Machine Learning.
>
> [4] Woo, Gerald, et al. "Unified Training of Universal Time Series Forecasting Transformers." Forty-first International Conference on Machine Learning.
>
> **Response to “Why BERT can have better performance than llama3"**
>
> We thank the reviewer for their insightful comments. Our clarification is as follows:
>
> 1. As shown in Figure 7(b), their performances are quite similar.
>
> 2. We have discussed the potential reasons in Lines 261-266, including: a) Our proposed simple multimodal framework still does not fully utilize the power of LLMs; b) The  capabilities of pretrained LLMs do not easily transfer to time-series tasks. This perspective has been recently validated by [1].
>
>  3. Additionally, BERT’s embedding dimension is 768, much lower than LLaMA3-8B’s 4096, making it easier to fintune an effective projection layer.
>
> We will include these extra discussions in the final version.
>
> [1] Tan, Mingtian, et al. "Are Language Models Actually Useful for Time Series Forecasting?." arXiv preprint arXiv:2406.16964 (2024).
>
> **Response to “Why is the horizon window size stable for baselines?  A longer window size could improve the performance"**
>
> We would like to respectfully correct the reviewer: the horizon window refers to the predicted series, not the input series. Therefore, a longer horizon does not provide additional information to improve performance. Our results in Figure 7(c) demonstrate that the introduction of text improves performance across different prediction lengths.
>
> **Response to Opportunities “Consider image and audio.”**
>
> We thank the reviewer for their constructive suggestions.
>
> In this work, we chose to include text for the following reasons: (1) Text and numerical series commonly coexist across multiple domains. (2) With the rise of LLMs, an increasing number of time-series methods are incorporating text based on LLM, although only predefined prompts [1-4]. Therefore, we constructed such a text-numerical multimodal dataset across 9 domains, which not only has significant real-world importance but also overcome current obstacles for method research.
>
> We agree with the reviewer that considering image and audio would be better. However, collecting data that aligns image, audio, and numerical modalities is quite challenging and constrained by domain selection, which is beyond the scope of this work. We will discuss the possibility and challenges of the mentioned modalities in the limitations section of the final version.
>
> [1] Jin, Ming, et al. "Time-LLM: Time Series Forecasting by Reprogramming Large Language Models." The Twelfth International Conference on Learning Representations. ICLR 2024.
>
> [2] Cao, Defu, et al. "TEMPO: Prompt-based Generative Pre-trained Transformer for Time Series Forecasting." The Twelfth International Conference on Learning Representations.
>
> [3] Xue, Hao, and Flora D. Salim. "Promptcast: A new prompt-based learning paradigm for time series forecasting." IEEE Transactions on Knowledge and Data Engineering (2023).
>
> [4] Gruver, Nate, et al. "Large language models are zero-shot time series forecasters." Advances in Neural Information Processing Systems 36 (2024).

---

> ### Author Rebuttal · Authors · 2024-08-17
>
> **Response to Opportunities “Consider multi-variables.”**
>
> We thank the reviewer for their constructive suggestions. We focused on the single variable in this work because ensuring high-quality text-numerical alignment is non-trivial. We require customized text data sources, data crawling and filtering targeted at the given variable. Thus, constructing a dataset for multiple variables requires building multiple aligned text series correspondingly, which needs a lot of effort.  We would also like to point out the single variables we selected are composite indicators, as detailed in Table 1. We plan to support multi-variables in the future. We will mention the lack of multi-variables in the limitations section of the final version.
>
> **Response to “They lack the discussion about the related work.”**
>
> We respectfully disagree with the reviewer. We would like to clarify that lines 38-51 and Appendix A **already introduced relationships with prior work**. We are willing to provide **a more comprehensive discussion in the form of a table**. We will incorporate these in the final version of our paper.

---

> ### Author Response · Authors · 2024-08-27
> **Kindly Request for Reviewer's Feedback**
>
> Dear Reviewer mAU8
>
> Since the End of author/reviewer discussions is very soon, may we know if our response addresses your main concerns? We will be more than happy to engage in more discussion and paper improvements.
>
> Thanks very much.

---

### Official Review · Reviewer_BWHX · 2024-07-24
**This paper introduces the Time-MMD dataset and MM-TSFlib framework for time series prediction, integrating textual and numerical data. Covering nine domains, Time-MMD ensures precise alignment and pollution removal.**

**Rating:** 8
**Confidence:** 5
**Clarity:** yes

**Review:**

This paper addresses the issues of sparse textual data, excessive data noise, and the difficulty of precise alignment in multi-domain multimodal time series prediction tasks by proposing efficient methods for numerical sequence construction, textual sequence construction, and numerical-textual sequence alignment. Consequently, a new multi-domain multimodal time series dataset has been successfully constructed. An innovative framework, MM-TSFlib, which integrates textual and numerical information, is also proposed. The construction of the Time-MMD dataset and the MM-TSFlib library is introduced with rigorous experimental design and detailed data processing procedures. Finally, the experimental results fully validate the effectiveness of the proposed methods, providing detailed results analysis and discussion.

Pros:

1.	Time-MMD is the first multimodal time series dataset covering multiple domains, addressing the deficiencies and poor quality of existing multimodal time series prediction datasets, and providing a rich data source.

2.	For the issues of sparse textual data, severe data pollution, and difficult alignment, an innovative method based on LLM for textual data screening and time step alignment is proposed.

3.	The MM-TSFlib framework, which combines textual and numerical information for time series prediction, is proposed and demonstrated through experiments to significantly improve time series prediction performance on the Time-MMD dataset.


Cons:

1.	The dataset has certain language limitations. Currently, the textual data is limited to English, lacking multilingual support, which may affect the applicability in non-English contexts.

2.	In the MM-TSFlib framework, textual information encoded by LLM is combined with numerical information only through fine-tuning the projection layer, failing to fully exploit the potential of textual information.

**Strengths:**

1.	Time-MMD is the first multimodal time series dataset encompassing multiple domains. By integrating numerical and textual data, this dataset addresses the shortcomings of existing datasets in terms of modality diversity and data quality, advancing the frontier of multimodal time series analysis.

2.	An innovative method based on LLM for textual data screening and time step alignment is proposed to address issues of sparse textual data, severe data pollution, and difficult alignment.

3.	The MM-TSFlib framework, which combines textual and numerical information for time series prediction, is introduced. It provides a seamless end-to-end pipeline, enabling researchers to easily extend multimodal applications to existing time series prediction models. Experiments demonstrate that using this framework on the Time-MMD dataset significantly improves time series prediction performance.

4.	The dataset supports a variety of time series prediction tasks across different fields. Time-MMD covers nine major data domains, including agriculture, climate, economy, energy, environment, health, safety, social welfare, and transportation. This reflects the dataset's broad applicability across multiple domains and scenarios, providing valuable data support for research in various fields.

5.	The open-source of Time-MMD facilitates the development of multimodal time series prediction tasks and promotes multimodal data fusion, which pave a new way for time series prediction.

**Additional Feedback:**

This dataset provides a diverse and effective resource for multimodal time series prediction tasks, with a detailed construction process that effectively addresses issues of sparse and polluted textual data. It is hoped that in the future, the dataset can be expanded to include multilingual textual data to enhance its global applicability. Additionally, developing a more automated data collection process will facilitate timely updates and maintainability of the

**Correctness:**

Yes,the claims made in the submission is corect. The dataset is constructed in a sound way.

**Documentation:**

Yes , there is sufficient detail on data collection and organization. The paper provides a detailed description of the data sources, selection criteria, and collection methods for the Time-MMD dataset. The data sources include government reports, news articles, and search results, ensuring diversity and reliability of the data. Additionally, open-source links and the MM-TSFlib library are provided, validating the effectiveness of this dataset.

**Limitations:**

The authors have addressed some limitations and potential negative social impact issues in this work. However, improvements and enhancements may still be needed in supporting language diversity and the maintenance and updating of the dataset.

**Opportunities For Improvement:**

1.	The current dataset's textual data is limited to English, lacking support for other languages. This limitation restricts the applicability of Time-MMD in non-English environments. Future improvements could include collecting and integrating multilingual textual data to enhance the dataset's adaptability to various languages.

2.	Merely fine-tuning the projection layer of the LLM may not fully integrate textual and numerical information. Future research should explore more effective methods for integrating these types of information.

**Relation To Prior Work:**

Yes

**Summary And Contributions:**

Addressing the issues of unimodal and heavily polluted multimodal data in time series prediction tasks, this paper introduces a new multi-domain multimodal time series prediction dataset, Time-MMD, and a multimodal time series prediction framework, MM-TSFlib, that integrates textual and numerical information. Time-MMD encompasses nine major data domains, with precise modality alignment and data pollution removal ensured through manual collection and LLM screening. Comparative experiments between MM-TSFlib and other unimodal time series prediction models on Time-MMD demonstrate a significant performance improvement by extending from unimodal numerical information to multimodal. The average Mean Squared Error (MSE) is reduced by more than 15%, and in some domains with rich textual information, the reduction is as high as 40%

---

> ### Author Rebuttal · Authors · 2024-08-17
>
> We thank the reviewer for their insightful comments and recognition.
>
> **Response to Concerns “lacking multilingual support”**
>
> We appreciate and agree with the reviewer's suggestion. We chose English to construct the first multi-domain multimodal time-series dataset, inspired by research experience in language models. For example, BERT[1] was originally trained on the English corpus only, and then expanded to multiple languages. Expanding our Time-MMD dataset to include multiple languages is indeed valuable. We have noted this limitation in Appendix B. However, such an expansion requires considerable extra effort thus is beyond the scope of this work. We plan to address this in future work.
>
> **Response to Concerns “ Merely fine-tuning the projection layer is not effective enough."**
>
> We appreciate and agree with the reviewer's suggestion. The goal of MM-TSFlib is to pioneeringly demonstrate a simple but effective usage of Time-MMD, which can guide further research using this multimodal dataset. We have mentioned the reviewer's concern in Appendix B. We envision that future work can build on our dataset, library, and experiments to explore advanced solutions for fine-tuning LLMs for multimodal TSA, but this is beyond the scope of this dataset work.
>
> **Response to  Limitations “updating of the dataset”**
>
> Our data sources all have reliable updates guaranteed, as detailed in Appendix C. We commit to maintaining and regularly updating this dataset and library.
>
> [1] https://github.com/google-research/bert

---

> > ### Comment · Reviewer_BWHX · 2024-08-21
> >
> > Thank you very much for your response. I have no further concerns

---

> > > ### Author Rebuttal · Authors · 2024-08-21
> > >
> > > We thank the reviewer for the recognition of our work. We are willing to answer additional questions, if any, after peer-reviewing the comments from other reviewers.

---

### Official Review · Reviewer_HR7m · 2024-07-26
**Review of Time-MMD**

**Rating:** 6
**Confidence:** 4
**Correctness:** Looks good to me.
**Clarity:** The paper is well-written in general.

**Review:**

See strengths and limitations.

**Strengths:**

- The paper is well-organized with pretty good presentations, and substantial details.
- I checked a few rounds and found this one to be the first dataset for text+time series multi-modal data. The data construction is non-trivial and requires quite a load of human labor.
- The experiments look good to me, having enough baseline models and comparing uni VS multi-modal performance on the forecasting task.
- The library would make it more practical and impactful.

**Additional Feedback:**

I think the dataset is valuable and I see the efforts behind this paper. The limitations more or less make the experiment results less exciting.

**Documentation:**

It looks like the github repo is incomplete and short of necessary instructions. If this can be done soon then it should be fine.

**Ethics:**

Looks good to me.

**Limitations:**

I do see the fact that given the large scale of textual data, LLM is the optimal choice, but I am still curious about the following two questions:
- (1) Does multi-modality with text data mean using LLM by default?
- (2) Is combining Open-source LLM output embeddings with the time series model embedding the only way of multi-modal time series modeling?

Though I do believe this paper shows good potential, but on the two aspects above, it is more or less hand-tied.

For (1), to the best of my knowledge the task of multi-modal TSA has not seen a comprehensive benchmark which compares various methods that may or may not use LLMs. There can also be many scenarios where the textual data is in small scale and won't bother using LLMs. The scope of this paper skips this necessary stage -- though I do agree with the uniqueness and value of the dataset.

For (2), it seems to be ad-hoc in this paper and the purpose is just to show results. It would be nicer if other proposed works are tested on this new dataset.

In short, the experiment results do not serve as a useful benchmark.

Another minor suggestion is to reduce the claim of significance on broader impact. It looks to me that any of the mentioned future directions is exciting, but really requires a lot more efforts and it is not even clear how they are related to this work on the technical details, beyond the multi-modality property itself.

**Opportunities For Improvement:**

See limitations

**Relation To Prior Work:**

It is not well-discussed. I would suggest adding some comparisons with prior data, maybe a table, depending on the details. The search space can be larger, I roughly searched and find a prior paper [1], though it is not text+TS but if there are more, it would be good to let people compare.

[1] WildfireSpreadTS: A dataset of multi-modal time series for wildfire spread prediction

**Summary And Contributions:**

This paper introduces a multimodal dataset for time series analysis, with the main motivation to exploit textual information in TSA. The dataset includes subsets collected from 9 different domains. A public library is provided for convenient use of the dataset, containing various popular time seires models. This library also enables an unimodal model to be converted to multi-modal by looping LLM embeddings into a pooling layer.

---

> ### Author Rebuttal · Authors · 2024-08-17
>
> We appreciate the reviewer's insightful comments and recognition of the value of our dataset and library. First of all, we would like to clarify that **research in multimodal TSA is still in its early stages**. The **main objective** of our dataset work is to **provide a high-quality multimodal dataset across various domains** to advance this field. Our library and experimental results serve as a pilot study to demonstrate the value of our dataset, the feasibility of multimodal TSA, and to inspire future method designs.
>
> We greatly appreciate the reviewer's questions, which made us realize that our initial exploration of method designs was not comprehensive enough. We will include the following discussion in the final version and have already **expanded our library to include the extra multimodal solutions** discussed below.
>
> **Response to “Do not serve as a useful benchmark result."**
>
> We appreciate the reviewer's insightful comments. The experiment results have verified the value of our dataset for time-series forecasting tasks; And it **cannot serve as a benchmark result at this time due to the early stage of research in this field**.
>
> We would like to clarify that previously proposed multimodal models [1-2] are not suitable for our dataset. The text used in these models is prompt, such as task description, rather than our exogenous textual series. Prompts are static, fixed in length, highly templated and do not provide additional information, thus differ significantly from textual series in our Time-MMD.
>
> We are still willing to conduct experiments using this kind of method, following the reviewer's suggestions. We used TimeLLM [1], an advanced proposed model in this category, as a representative. The experimental results are as follows:
>
> | Model                     | Health | Energy | Traffic |
> |---------------------------|--------|--------|---------|
> | TimeLLM(Uni)              | 1.92  | 0.30 | 0.23   |
> | TimeLLM(Multi_TimeLLM)    | 2.51   | 0.39   | 0.27   |
> | TimeLLM(Multi_Our)        | 1.42   | 0.25   | 0.22   |
>
>
> The results demonstrate that the "prompt as prefix" approach in TimeLLM, denoted as Multi_TimeLLM,  is not suitable for textual series. Possible reasons include the longer length of text series might overwhelm the numerical series information, among others. However, our framework, although simple, is effective.
>
> [1] Jin, Ming, et al. "Time-LLM: Time Series Forecasting by Reprogramming Large Language Models." The Twelfth International Conference on Learning Representations. ICLR 2024.
>
> [2] Cao, Defu, et al. "TEMPO: Prompt-based Generative Pre-trained Transformer for Time Series Forecasting." The Twelfth International Conference on Learning Representations.

---

> > ### Author Rebuttal · Authors · 2024-08-17
> >
> > **Response to “Does multi-modality with text data mean using LLM by default?”**
> >
> > We appreciate the reviewer's insightful question. We have demonstrated the value of using LLMs, which are currently considered the mainstream method for text modeling. And, we agree with the reviewer that there are also many scenarios where LLMs may not be suitable. Therefore, we are willing to **introduce Doc2Vec [1], a embedding model trained from scratch**, for text modeling. Our experimental results on three datasets across four prediction lengths are as follows.
> >
> >
> > | Dataset           | Health | Energy | Traffic |
> > |-------------------|--------|--------|---------|
> > | Reformer(Uni)    | 1.85   | 0.43   | 0.31    |
> > | Reformer(Multi_Bert) | 1.27   | 0.40   | 0.17    |
> > | Reformer(Multi_Doc2Vec) | 1.35   | 0.43   | 0.22    |
> > | Informer(Uni)     | 1.53   | 0.33   | 0.28    |
> > | Informer(Multi_Bert) | 1.22   | 0.29   | 0.17    |
> > | Informer(Multi_Doc2Vec) | 1.32   | 0.31   | 0.21    |
> >
> >
> > We can find that Doc2Vec is also effective, although generally performed weaker than Bert. Doc2Vec is now **included in our library**, which will greatly enhance its efficiency and applicability to underrepresented languages and domains. Once again, we thank the reviewer for this inspiring question!
> >
> > [1] Le, Quoc, and Tomas Mikolov. "Distributed representations of sentences and documents." International conference on machine learning. PMLR, 2014.
> >
> > **Response to “Is combining the only way of multi-modal time series modeling?”**
> >
> > We appreciate the reviewer's insightful question.  We agree that simply combining is not the only approach for multimodal TSA. Although our dataset work does not aim to explore the optimal method, we are willing to introduce **three additional solutions** to inspire future research, which are now **integrated into our library**.
> >
> > 1. From "combining" to "attention" :
> >
> > We use the output of the time series model to calculate attention scores for each token in the LLM output and perform weighted aggregation. For detailed implementation, please refer to our library. The experimental results are as follows:
> >
> > | Dataset               | Health | Energy | Traffic |
> > |-----------------------|--------|--------|---------|
> > | Reformer(Multi_Comb) | 1.27   | 0.41   | 0.17    |
> > | Reformer(Multi_Att)  | 1.26   | 0.39   | 0.22    |
> > | Informer(Multi_Comb)  | 1.22   | 0.29   | 0.17    |
> > | Informer(Multi_Att)   | 1.23   | 0.29   | 0.16    |
> >
> > This basic attention method has not yet demonstrated a clear advantage over the current simple combing approach. We look forward to future method designs.
> >
> > 2. From "OpenSource LLM" to "small models trained from scratch":
> >
> > As discussed above, we have also verified that Doc2Vec is effective.
> >
> > 3. from "OpenSource LLM" to "ClosedSource LLM’
> >
> > We first used ClosedSource LLM, such as GPT-3.5, to generate text-based predictions, and then encoded them using Bert. For detailed implementation, please check our library. The experimental results are as follows:
> >
> > | Dataset               | Health | Energy | Traffic |
> > |-----------------------|--------|--------|---------|
> > | Reformer(Uni)         | 1.86   | 0.43   | 0.31    |
> > | Reformer(Multi_Bert)  | 1.27   | 0.40   | 0.17    |
> > | Reformer(Multi_GPT3.5)| 1.38   | 0.46   | 0.19    |
> > | Informer(Uni)         | 1.53   | 0.33   | 0.28    |
> > | Informer(Multi_Bert)  | 1.22   | 0.28   | 0.17    |
> > | Informer(Multi_GPT3.5)| 1.33   | 0.32   | 0.23    |
> >
> >
> >
> >
> > We initially observed that fully frozen closed-source LLM are less effective than fine-tuned open-source LLM.
> >
> >
> > If the reviewer is interested in seeing more models and experiments, we would be more than happy to provide them. Once again, we appreciate the reviewer for raising these two thought-provoking questions, even though they go beyond the scope of this dataset work.

---

> > > ### Author Rebuttal · Authors · 2024-08-17
> > >
> > > **Response to “reduce the claim of significance on broader impact”**
> > >
> > > We appreciate the reviewer's constructive suggestions. Regarding the three aspects mentioned in the broader impact section, we can currently directly support multimodal imputation tasks; please check at dataset repo. We have also provided an approach for multimodal anomaly detection and an initial labeled dataset, available at https://anonymous.4open.science/r/Time-MMD-Anomaly-9424/. We plan to release this publicly after manual verification.
> > >
> > >
> > > We are also willing to revise the claim. In the final version, we will change Section 5 from "Broader Impacts" to "Potential Future Work," and include more specific implementation details and challenges.
> > >
> > >
> > > **Response to “Relation to prior work is not well-discussed”**
> > >
> > > We appreciate the reviewer's constructive suggestions. We would like to clarify that lines 38-51 and Appendix A already introduced relationships with previous text + numerical multimodal data. The image + text multimodal data, mentioned by the reviewer, primarily serves spatial-temporal data analysis, rather than our focused time-series data analysis.
> > >
> > > We are willing to provide **a more comprehensive discussion in the form of a table**, including additional multimodal types as suggested by the reviewer. Please check the **attached PDF**. We will incorporate these in the final version of our paper.
> > >
> > >
> > > **Response to "Documentations need improvement"**
> > >
> > > We appreciate the reviewer's constructive suggestions. We have **significantly improved the documentations** including detailed description, usage guidelines, and news list. We have also expanded the functionality of the dataset and library. Please check the **updated** dataset at https://github.com/AdityaLab/Time-MMD   and library at https://github.com/AdityaLab/MM-TSFlib

---

> > > > ### Comment · Reviewer_HR7m · 2024-08-22
> > > >
> > > > Thank the authors for conducting additional experiments. The results make sense to me, for combining VS attention, it is not surprising (but disappointing) that they give similar results. Though the insights are limited, this is mostly because of lack of well-developed methods for multi-modal time series modeling, and out of the scope of this paper.
> > > >
> > > > For the anomaly detection and long-term forecasting task, I appreciate the efforts. What model did you use exactly?
> > > >
> > > > I could raise the score to 6 and vote for acceptance.

---

> > > > > ### Author Rebuttal · Authors · 2024-08-24
> > > > >
> > > > > We greatly appreciate the reviewer's recognition of our work and the insightful questions for future work. We have utilized over 20 different time series models, which are generally applicable for long-term forecasting and anomaly detection tasks. Detailed model list is at https://github.com/AdityaLab/MM-TSFlib/tree/main/models.

---

### Author Rebuttal · Authors · 2024-08-17

**General Rebuttal**

We appreciate the reviewers' recognition of the value of our **dataset, library and experiments**. Multimodal time series analysis (TSA) is still in its early research stages due to the lack of suitable datasets. Existing datasets suffer from several limitations, including narrow data domains (focused on finance), coarse-grained modality alignment (aligning general finance news with a specific stock), and inherent data contamination (the entanglement of objective facts with subjective predictive text). By addressing these issues, Time-MMD serves as **the first high-quality multimodal dataset across 9 domains**, dedicated to overcoming the current major obstacle. Our library and experimental results serve as **a pioneering study in multimodal time series forecasting, validating the value of the dataset and inspiring future work**. We commit to continuously maintaining and updating the dataset and library.

We appreciate the reviewers' insightful comments and constructive suggestions. Thus, we have made the following **improvements**:

1. **More Clear Documentation:** We have improved the usage documentation to ensure usability. We have provided detailed description, usage guidelines, and news list.

2. **Expanded Supported Tasks:** The dataset now supports four time series tasks. It supports the most common short-term forecasting, long-term forecasting and additional imputation and anomaly detection tasks. For imputation, we generate missing values following mainstream methods, released at our repo.
For anomaly detection, we have provided preliminary anomaly labels, available at https://anonymous.4open.science/r/Time-MMD-Anomaly-9424/
We plan to release it after manual verification.

3. **Extended Library Functionality:** The library now supports various open-source LLMs, closed-source LLMs, and a Doc2Vec model trained from scratch as text modeling approaches. Additionally, we have included multiple multimodal fusion solutions, including attention mechanisms.

4. **Extra Experiments and Discussion:** For the newly added multimodal solutions, we have conducted experiments and provided discussions, detailed in responses to reviewer HR7m. These will be included in the final version.

5. **More Comprehensive Related Work Discussion:** In the attached PDF, we have provided a more comprehensive comparison of existing dataset works in tabular form. We will include this in the final version.

We will also address some writing issues in the final version, such as reducing the emphasis on broader impact and adding extra discussion on limitations.

Please check the **updated** dataset at https://github.com/AdityaLab/Time-MMD   and library at https://github.com/AdityaLab/MM-TSFlib

---

### Decision · Program_Chairs · 2024-09-26

**Decision:**

Accept (Poster)

**Comment:**

The paper introduces the first benchmark for time-series forecasting and analysis that is also multimodal, that is, it also contains related texts that have to be taken into account to model the temporal data accurately. All reviewers are positive about the paper, although the authors should look into making the broader impact statement more in line with the actual content. I also went through the paper, and I think it is a valuable resource, so it is fair that it will be accepted.

Out of curiosity, perhaps it would be interesting to include a discussion on how this benchmark could relate to (spatio)temporal and/or interacting dynamical systems and equation discovery, where perhaps language could inform of the qualitative behavior of the system, eg in https://icml.cc/virtual/2024/poster/32807.